



# Ten years of atmospheric methane from ground-based NDACC FTIR observations

Whitney Bader[1,2], Benoît Bovy[1], Stephanie Conway[2], Kimberly Strong[2], Dan Smale[3], Alexander J. Turner[4], Thomas Blumenstock[5], Chris Boone[6], Ancelin Coulon[7], Omaira Garcia[8], David W. T. Griffith[9], Frank Hase[5], Petra Hausmann[10], Nicholas Jones[9], Paul Krummel[11], Isao Murata[12], I. Morino[13], Hideaki Nakajima[13], Simon O'Doherty[14], Clare Paton-Walsh[9], John Robinson[3], Rodrigue Sandrin[2], Matthias Schneider[5], Christian Servais[1], Ralf Sussmann[10] and Emmanuel Mahieu[1]

[1]Institute of Astrophysics and Geophysics, University of Liège, Liège, Belgium
[2]Department of Physics, University of Toronto, Toronto, ON, M5S 1A7, Canada
[3]National Institute of Water and Atmospheric Research, NIWA, Lauder, New Zealand
[4]School of Engineering and Applied Sciences, Harvard University, Cambridge, MA, USA
[5]Karlsruhe Institute of Technology (KIT), Institute of Meteorology and Climate Research (IMK-ASF), Karlsruhe, Germany
[6]Department of Chemistry, University of Waterloo, Waterloo, ON, N2L 3G1, Canada
[7]Institute for Atmospheric and Climate Science, ETH Zurich, Zurich, Switzerland
[8]Izana Atmospheric Research Centre (IARC), Agencia Estatal de Meteorologia (AEMET), Spain
[9]School of Chemistry, University of Wollongong, Australia
[10]Karlsruhe Institute of Technology, IMK-IFU, Garmisch-Partenkirchen, Germany
[11]CSIRO Oceans & Atmosphere, Aspendale, Victoria, Australia
[12]Graduate School of Environment Studies, Tohoku University, Sendai 980-8578, Japan
[13]National Institute for Environmental Studies (NIES), Tsukuba, Ibaraki 305-8506, Japan
[14]Atmospheric Chemistry Research Group (ACRG), School of Chemistry, University of Bristol, UK

*Correspondence to*: Whitney Bader (wbader@atmosp.physics.utoronto.ca)

**Abstract**

Changes of atmospheric methane ($CH_4$) since 2005 have been evaluated using Fourier Transform Infrared (FTIR) solar observations performed at ten ground-based sites, all members of the Network for Detection of Atmospheric Composition Change (NDACC). From this, we find an increase of atmospheric methane total columns that amounts to $0.31 \pm 0.03$ % year$^{-1}$ (2-sigma level of uncertainty) for the 2005-2014 period. Comparisons with in situ methane measurements at both local and global scales show good agreement. We used the GEOS-Chem Chemical Transport Model tagged simulation that accounts for the contribution of each emission source and one sink in the total methane, simulated over the 2005-2012 time period and based on emissions inventories and transport. After regridding according to NDACC vertical layering using a conservative regridding scheme and smoothing by convolving with respective FTIR seasonal averaging kernels, the GEOS-Chem simulation shows an increase of atmospheric methane of $0.35 \pm 0.03$ % year$^{-1}$ between 2005 and 2012, which is in agreement with NDACC measurements over the same time period ($0.30 \pm 0.04$ % year$^{-1}$, averaged over ten stations). Analysis of the GEOS-Chem tagged simulation allows us to quantify the contribution of each tracer to the global methane change since 2005. We find that natural sources such as wetlands and biomass burning contribute to the inter-annual variability of methane. However, anthropogenic emissions such as coal mining, and gas and oil transport and exploration, which are mainly emitted in the Northern Hemisphere and act as secondary contributors to the global budget of methane, have played a major role in the increase of atmospheric methane observed since 2005. Based on the GEOS-Chem tagged simulation, we discuss possible cause(s) for the increase of methane since 2005, which is still unexplained.





## 1 Introduction

Atmospheric methane ($CH_4$), a relatively long-lived atmospheric species with a lifetime of 8-10 years (Kirschke et al., 2013), is the second most abundant anthropogenic greenhouse gas, with a radiative forcing (RF) of $0.97 \pm 0.23$ W m$^{-2}$ (including indirect radiative forcing associated with the production of tropospheric ozone and stratospheric water vapor), after $CO_2$ (RF

in 2011: $1.68 \pm 0.35$ W m$^{-2}$, Stocker et al., 2013). Approximately one-fifth of the increase in radiative forcing by human-linked greenhouse gases since 1750 is due to methane (Nisbet et al., 2014). Identified emission sources include anthropogenic and natural contributions. Human activities associated with the agricultural and the energy sectors are the main sources of anthropogenic methane through enteric fermentation of livestock (17 %), rice cultivation (7 %), for the former, and coal mining (7 %), oil and gas exploitation (12 %), and waste management (11 %), for the latter. On the other hand, natural sources of

methane include wetlands (34 %), termites (4 %), methane hydrates and ocean (3 %) along with biomass burning (4 %), a source of atmospheric methane that is both natural and anthropogenic. The above-mentioned estimated contributions to the atmospheric content of methane are based on Chen and Prinn (2006), Fung et al. (1991), Kirschke et al. (2013) and on emission inventories used for the GEOS-Chem v9-02 methane simulation (Turner et al., 2015), and it is worth noting that these figures are still affected by significant uncertainties.

Methane is depleted at the surface by consumption by soil bacteria, in the marine boundary layer by reaction with chlorine atoms, in the troposphere by oxidation with the hydroxyl radical (OH), and in the stratosphere by reaction with chlorine atoms, O($^1$D), OH, and by photodissociation (Kirschke et al., 2013). Due to its sinks, methane has important chemical impacts on the atmospheric composition. In the troposphere, oxidation of methane is a major regulator of OH (Lelieveld, 2002) and is a source

of hydrogen and of tropospheric ozone precursors such as formaldehyde and carbon monoxide (Montzka et al., 2011). In the stratosphere, methane plays a central role as a sink for chlorine atoms and as a source of stratospheric water vapor, an important driver of decadal global surface climate change (Solomon et al., 2010). Given its atmospheric lifetime, its impact on radiative forcing and on atmospheric chemistry, methane is one of the primary targets for regulation of greenhouse gas emissions and climate change mitigation.

As a result of growing anthropogenic emissions, atmospheric methane showed prolonged periods of increase over the past three decades (World Meteorological Organization, 2014). From the 1980s until the beginning of the 1990s, atmospheric methane was rising sharply by about ~ 0.7 % year$^{-1}$ (Nisbet et al., 2014) but stabilized during the 1999-2006 time period (Dlugokencky, 2003). Many studies were dedicated to the analysis of methane trends, in particular the stabilization of methane

concentrations between 1999 and 2006, and various scenarios have been suggested. They include reduced global fossil-fuel-related emissions (Aydin et al., 2011; Chen and Prinn, 2006; Simpson et al., 2012; Wang et al., 2004), a compensation between increasing anthropogenic emissions and decreasing wetland emissions (Bousquet et al., 2006), and/or significant (Rigby et al., 2008) to small (Montzka et al., 2011) changes in OH concentrations. However, Pison et al. (2013) emphasized the need for a comprehensive and precisely quantified methane budget for its proper closure and the development of realistic future climate

scenarios.

Since 2005-2006, a renewed increase of atmospheric methane has been observed and widely discussed in many studies (Bloom et al., 2010; Dlugokencky et al., 2009; Frankenberg et al., 2011; Hausmann et al., 2016; Helmig et al., 2016; Montzka et al., 2011; Rigby et al., 2008; Schaefer et al., 2016; Spahni et al., 2011; Sussmann et al., 2012; van der Werf et al., 2010), leading

to various hypotheses. In this work, for the first time, we report of an increase of methane as observed since 2005 at a suite of NDACC sites distributed worldwide and operating Fourier Transform InfraRed (FTIR) Spectrometers. The paper is organized as follows: Sect. 2 includes a brief description of the ten participating sites, and the retrieval strategy and information content




of the FTIR measurements. Sect. 3 focuses on the methane changes since 2005 as derived from the NDACC FTIR measurements, and the GEOS-Chem model, along with comparisons between both model and observations. This section also provides a source-oriented analysis of the recent increase of methane using the GEOS-Chem tagged simulation. Finally, Sect. 4 discusses the potential source(s) responsible for the observed increase of methane since the mid-2000s.

## 2 NDACC FTIR observations

The international Network for the Detection of Atmospheric Composition Change (NDACC) is dedicated to observing and understanding of the physical and chemical state of the stratosphere and troposphere. Its priorities include the detection of trends in atmospheric composition, understanding their impacts on the stratosphere and troposphere, and establishing links between climate change and atmospheric composition.

### 2.1 Observation sites

Ground-based NDACC FTIR measurements of methane obtained at ten globally distributed observation sites are presented in this study. These sites, displayed on Fig. 1 and whose location is detailed in Table 1 are located from North to South in Eureka (Arctic, Canada), Kiruna (Sweden), Zugspitze (Germany), Jungfraujoch (Switzerland), Toronto (Canada), Tsukuba (Japan), Izaña (Canary Island, Spain), Wollongong (Australia), Lauder (New Zealand), Arrival Heights (Antarctica). Most of the FTIR data is available on the NDACC database (http://www.ndsc.ncep.noaa.gov/data/).

The Eureka (EUR, Fogal et al., 2013) station is located in the Canadian high Arctic, at 610 m a.s.l. on Ellesmere Island in the northern Canadian Archipelago. The station is located along the Slidre Fjord and is surrounded by complex topography (Cox et al., 2012). This topography, along with its proximity to the Greenland Ice Sheet and atmospheric conditions, make this station ideal for infrared solar measurements in the Arctic as it is frequently under the influence of cold and dry air from central Arctic and the Greenland Ice Sheet (Cox et al., 2012). Routine solar infrared measurements are performed from late February to late October), no lunar measurements are taken during polar night (Batchelor et al., 2009).

The Kiruna (KIR) site is located in the boreal forest region of Northern Sweden. The spectrometer is operated in the building of the IRF (Institute för Rymdfysik/Swedish Institute of Space Physics), at an altitude of 420 m, about 10 km away from the center of Kiruna, the northernmost town of Sweden. The local population and traffic density is low, so the FTIR site is not significantly affected by local anthropogenic emissions. The location just inside the polar circle is especially suited for the study of the Arctic polar stratosphere, because the break in solar absorption observations is still rather short, while the stratospheric polar vortex frequently covers Kiruna in early spring. The solar absorption spectra were obtained with a Bruker IFS-120HR since 1996. In 2007, an electronic upgrade to a Bruker IFS-125HR was implemented. Routine solar infrared measurements are performed between mid-January and mid-November, no lunar measurements are taken during polar night.

The Zugspitze site (ZUG, Sussmann and Schäfer, 1997) is located on the southern slope of the Zugspitze mountain, the highest mountain of the German Alps (2964 m a.s.l.), at the Austrian border near the town of Garmisch-Partenkirchen (720 m a.s.l.). Its high-altitude offers an excellent location for long-term trace gas measurements under unperturbed background atmospheric conditions and exhibits a very low level of integrated water vapor.

The Jungfraujoch (JFJ, Zander et al., 2008) station is located in the Swiss Alps at 3580 m altitude on the saddle between the Jungfrau (4158 m a.s.l.) and the Mönch (4107 m a.s.l.) summits. This station offers unique conditions for infrared solar observations because of weak local pollution (no major industries within 20 km) and very high dryness due to the high-altitude





and the presence of the Aletsch Glacier in its immediate vicinity. The Jungfraujoch station allows the investigation of the atmospheric background conditions over central Europe and the mixing of air masses between the planetary boundary layer and the free troposphere (Reimann, 2004).

The Toronto (TOR) station is located in the core of the megacity of Toronto, Ontario, Canada at 174 m a.s.l. where regular solar measurements began in 2002. In contrast to most NDACC stations, the Toronto station is highly affected by the densely populated areas of the city of Toronto itself (the center of Canada's largest population) and the cities and industrial centers of the North Eastern United States, enabling measurements of tropospheric pollutants (Whaley et al., 2015). In addition, the station's location makes it well suited for measurements of mid-latitude stratospheric ozone, related species, and greenhouse

gases (Wiacek et al., 2007).

The Tsukuba (TSU) station is located in a suburban area (around 50 km from Tokyo) and in a large plain with many rice paddies at an altitude of 31 m. The station occasionally captures local pollution and is affected by high humidity during the summer season. The Tsukuba solar absorption spectra were obtained with a Bruker IFS-120HR from May 2001 to March

2010, and replaced by a Bruker IFS-125HR in April 2010.

The Izaña observatory (IZA, http://www.izana.org), is located on the top of a mountain plateau in the Teide National Park on the Island of Tenerife. It is usually located above a strong subtropical temperature inversion layer (generally well established between 500 and 1500 m a.s.l.) and clean-air and clear-sky conditions are prevailing year-round. Consequently it offers

excellent conditions for the remote sensing of trace gases and aerosols under "free troposphere" conditions and for atmospheric observations. Due to its geographic location, it is particularly valuable for the investigation of dust transport from Africa to the North Atlantic, and large-scale transport from the tropics to higher latitudes. In addition, during daytime the strong insolation generates a slight upslope flow of air originating from below the inversion layer (from a woodland that surrounds the station at a lower altitude, Sepúlveda et al., 2012). The solar absorption spectra were obtained with a Bruker IFS 120M over 1999-

2004, then with a Bruker IFS 125HR thereafter (Sepúlveda et al., 2012).

Wollongong (WOL, Griffith et al., 1998) is a coastal city about 80 km south of the metropolis of Sydney. Its urban location, in proximity to Sydney and local coal mining operations means that enhanced levels of $CH_4$ are measured from time to time. Climatologically the winds are weak (<4 m s-1); during the Southern Hemisphere winter the site largely samples continental

airmasses from the west, with summer afternoon sea-breezes from the East-Northeast (Fraser et al., 2011). The solar absorption spectra were obtained with a Bomem DA8 from 1995 to 2007 (Griffith et al., 1998) and with a Bruker IFS 125/HR from 2007 onwards.

The Lauder (LAU) atmospheric research station is located in the Manuherikia valley, Central Otago, New Zealand. The site

experiences a continental climate of hot dry summers and cool winters with a predominating westerly wind. The site is sparsely populated and remote from any major industries with non-intense agricultural and horticulture as the mainstay of economic activity.

The Arrival Heights (AHT) atmospheric laboratory is located 3 km north of McMurdo and Scott Base stations on Hut Point

Peninsula, the southern volcanic peninsula of Ross Island. With minimal exposure to local anthropogenic pollution and sources, methane measurements conducted at Arrival Heights are representative of a well-mixed boundary layer and free troposphere. Located at 78°S, Arrival Heights is periodically underneath the polar vortex depending on the season, polar vortex





shape and angular rotation velocity. Climatological surface meteorological conditions experienced at Arrival Heights are similar to that of Scott Base (Turner et al., 2004). Routine solar infrared measurements are performed during the austral spring and summer seasons (late August to mid-April) no measurements are taken during polar night.

### 2.2 FTIR Observations of methane

#### 2.2.1 Retrieval strategies

A retrieval strategy for the inversion of atmospheric methane time series from ground-based FTIR observations has been carefully developed and optimized for each station. However, it is worth mentioning that given the remaining inconsistencies affecting the methane spectroscopic parameters, even in the latest editions of HITRAN, the harmonization of retrieval strategies for methane for the whole infrared working group of NDACC is still ongoing. To this day, FTIR measurements are

analyzed as recommended either by Rinsland et al. (2006), Sussmann et al. (2011), or Sepúlveda et al. (2012). Table 2 presents the retrieval parameters used for each station. The retrieval codes PROFFIT (Hase, 2000) and SFIT-2/SFIT-4 (Rinsland et al., 1998) have been shown to provide consistent results for tropospheric and stratospheric species (Duchatelet et al., 2010; Hase et al., 2004). The time series produced using the strategies described in Table 2 are illustrated in Fig. 2. The various panels show daily mean methane time series expressed as anomalies with respect to a reference column in 2005.0 (2006.0 for the

Eureka station), as derived from the linear component fitted to the time series (see Sect. 3). The reference columns are given for each station in Table 3. It should be mentioned that the Toronto methane columns from 2008 to early 2009 present a systematic error due to an unknown instrument artifact. The dataset was corrected by adding a constant offset to the data over that period. To do this, a linear regression was first performed on the full dataset (20 June 2002 to 13 December 2014) excluding the biased data, and then another linear fit was performed only on the biased data (1 January 2008 to 19 March 2009) using

the same fixed slope. The difference between the two intercepts gives a constant offset of molecules cm$^{-2}$ that was added to the biased data.

In order to characterize the possible impact of the choice of the microwindows and spectroscopy on the retrieved methane, each strategy has been tested over a set of spectra recorded at the Jungfraujoch station (3068 spectra recorded between

01-01-2005 and 12-31-2012). Mean fractional differences between the strategies described in Table 2 have been computed to quantify a potential absolute bias in terms of total columns and changes over the 2005-2012 time period with the inversion strategy optimized for the Jungfraujoch observations set as a reference. Mean fractional differences are defined as the difference between two datasets divided by their arithmetic average and expressed in percent (see Eq. 2 in Strong et al., 2008). This results in an averaged bias between total columns of $0.9 \pm 0.5$ % but no bias between their respective trends since 2005

is observed (reference values associated with the JFJ strategy in Table 2 are a mean total column of $2.4121 \pm 0.0055 \times 10^{19}$ molecules cm$^{-2}$ and a mean annual change of $0.22 \pm 0.04$ % year$^{-1}$ with respect to 2005.0).

#### 2.2.2 Information content

Due to the previously mentioned unresolved discrepancies associated with methane spectroscopic parameters, it has been established within the NDACC Infrared Working Group that the regularization strength of the methane retrieval strategy should

be optimized so that the Degrees of Freedom For Signal (DOFS) is limited to a value of approximately 2 (Sussmann et al., 2011). As a consequence, the typical information content of NDACC methane retrievals will allow us to retrieve tropospheric and stratospheric columns, as displayed in Fig. 3. Indeed, the first eigenvector (in green) and its associated eigenvalue (typically close to 1) show that the corresponding information is mainly coming from the retrieval (> 99 %), allowing us to retrieve a partial column ranging from the surface up until 30 km. In addition, the second eigenvector allows for a finer vertical resolution

with two supplementary partial columns with typically around 16 % of a priori dependence: (i) a tropospheric column (typically





from the surface to the vicinity of the mean tropopause height of the station) along with (ii) a stratospheric column (from around the mean tropopause height to 30 km). In terms of error budget, extensive error analysis has been performed by Sepúlveda et al. (2014) and Sussmann et al. (2011). It has been determined that spectroscopic parameters almost exclusively determine the systematic error and amounts to ~2.5 % while statistical errors, dominated by baseline uncertainties and
measurement noise, sum up to ~1 % (Sepúlveda et al., 2014).

As illustrated in Fig. 3, the information content of our retrievals sets the upper and lower limit of respectively our tropospheric and stratospheric column at the vicinity of the mean tropopause height of the station. Therefore, the typical vertical sensitivity range of our retrieval restricts our definition of a purely tropospheric component. Indeed, our tropospheric column as previously
defined may potentially include a stratospheric contribution due to tropopause altitude variation, hence preventing the sampling of the free tropospheric column in some cases (Sepúlveda et al., 2014).

## 3. Methane changes since 2005

We characterize the global increase of methane total column from ten NDACC stations since 2005 and over 10 years' worth of observations, with a mean annual growth ranging from $0.26 \pm 0.02$ (Wollongong, 2-sigma level of uncertainty) to
$0.39 \pm 0.09$ % year$^{-1}$ (Toronto). Observational methane time series anomaly and their changes (along with their associated uncertainties) since 2005.0, illustrated in green in Fig. 4 and detailed in Table 3, have been analyzed for all ten sites using the statistical bootstrap resampling tool accounting for a linear component and a Fourier series taking into account the intra-annual variability of the dataset (Gardiner et al., 2008). As in Mahieu et al. (2014), the order of the Fourier series is adapted to each dataset depending on their sampling, i.e. limiting the order for the polar sites for which only a partial representation of the
seasonality is available. Anomalies of methane total column time series, illustrated in Fig. 4, have been computed using the methane total column computed by the linear component of the statistical bootstrap tool on 1 January 2005, as a reference. Table 3 shows trends of methane total column computed from FTIR observations over the 2005-2014 and 2005-2012 time periods as well as from a tagged GEOS-Chem simulation between 2005 and 2012. The latter is further discussed in Sect. 3.1.2.

On a regional scale, we compared our results with annual changes of methane as computed over the 2005-2014 time period from surface GC-MD observations (Gas Chromatography – Multi Detector) performed in the framework of the AGAGE program (Advanced Global Atmospheric Gases Experiment, Prinn et al., 2000) and from in situ surface measurements performed in the framework of the NOAA (National Oceanic and Atmospheric Administration) ESRL (Earth System Research Laboratory) carbon cycle air sampling network (Dlugokencky et al., 2015). Five representative observation sites have been
considered : Alert (Nunavut, Canada, 82.45 °N, -62.51 °E, 200.00 m a.s.l., Dlugokencky et al., 2015), Mace Head (Ireland, 53.33 °N, -9.90 °E, 5.00 m a.s.l., Prinn et al., 2000), Izaña (28.29 °N, 16.48 °W, 2372.90 m a.s.l., Dlugokencky et al., 2015), Cape Grim (Australia, 40.68 °S, 144.69 °E, 94.00 m a. s. l., Prinn et al., 2000), and Halley (United Kingdom, 75.61 °S, 26.21 °W, 30.00 m a.s.l., Dlugokencky et al., 2015).

Firstly, in situ measurements collected at Alert, representative of the northern polar region, show an increase of methane of $0.29 \pm 0.02$ % year$^{-1}$ (or $5.40 \pm 0.41$ ppb year$^{-1}$) since 2006 which is in agreement with our FTIR observations at Eureka with a mean annual change of $0.28 \pm 0.05$ % year$^{-1}$. For the northern mid-latitudes, we find an agreement between changes of methane as computed from surface measurements at Mace Head with an increase of $0.30 \pm 0.02$ % year$^{-1}$ (or $5.58 \pm 0.32$ ppb year$^{-1}$) and from our FTIR observations. Indeed, we observe consistent increases of methane of $0.32 \pm 0.03$,
$0.27 \pm 0.03$, and $0.29 \pm 0.08$ % year$^{-1}$ since 2005 at Zugspitze, Jungfraujoch and Toronto, respectively. Comparisons between changes of methane from FTIR and in situ surface measurements have also been performed for the Izaña station and show a





close to statistical agreement with respectively a mean annual increase of $0.33 \pm 0.01$ and $0.28 \pm 0.02$ % year$^{-1}$. In the southern hemisphere, AGAGE GC-MD measurements of methane at Cape Grim, representative of the mid-latitudes, shows a mean annual increase of $0.31 \pm 0.01$ % year$^{-1}$ (or $5.40 \pm 0.16$ ppb year$^{-1}$) which is in agreement with FTIR changes at Lauder of $0.29 \pm 0.03$ % year$^{-1}$. However, we should note the slightly larger mean annual changes of methane of Cape Grim in situ

observations with respect to Wollongong FTIR measurements. Indeed, it needs to be mentioned that FTIR measurements before the instrument change in 2007 (Bomem DA8 vs Bruker IFS 125HR, see Table 1) show noisier results. These noisier observations at the beginning of the time period under investigation may affect the relatively small annual changes of methane overall. As a result, the 2005-2007 time series shows no changes of methane while the 2007-2014 time period shows a mean annual change of $0.32 \pm 0.03$ % year$^{-1}$ (or $11.94 \pm 1.03$ x $10^{16}$ molecules cm$^{-2}$ year$^{-1}$) with respect to 2007.0, which is in

agreement with both Lauder FTIR and Cape Grim GC-MD methane changes since 2005. Finally, we computed a mean annual change of methane of $0.32 \pm 0.01$ % year$^{-1}$ (or $5.45 \pm 0.14$ ppb year$^{-1}$) from in situ surface measurements performed at Halley which is in good agreement with the mean annual change of methane computed from FTIR Arrival Heights retrievals that amounts at $0.32 \pm 0.07$ % year$^{-1}$.

In summary, we globally observe from NDACC FTIR measurements an average annual change of methane of $0.31 \pm 0.03$ % year$^{-1}$ (averaged over ten stations, 2-sigma level of uncertainty) which is in agreement with a mean annual change of $0.31 \pm 0.01$ % year$^{-1}$ (or $5.51 \pm 0.17$ ppb year$^{-1}$), as computed from the monthly global means of baseline data derived from AGAGE measurements (Prinn et al., 2000).

In addition, analysis of tropospheric and stratospheric partial columns changes show tropospheric mean annual changes of methane statistically in agreement (at the 2-σ level) with changes of total column over the 2005-2014 time period. Mean annual changes from the Atmospheric Chemistry Experiment-Fourier Transform Spectrometer methane research product (ACE-FTS, Bernath et al., 2005) have also been examined. For consistent comparison, ACE-FTS stratospheric columns of methane have been defined the same way than the stratospheric FTIR product, i.e. from the average tropopause height of the station to 30 km.

Changes of stratospheric methane according to ACE-FTS retrievals are statistically in agreement with our NDACC FTIR changes of stratospheric columns and show small to non-significant changes of methane in the stratosphere. Indeed, changes of stratospheric methane according to the ACE-FTS methane research product (Buzan et al., 2016) are not significant and amount to $-0.12 \pm 0.13$ % year$^{-1}$ for the northern high-latitudes, $0.10 \pm 0.30$ for northern mid-latitudes, $0.08 \pm 0.24$ for the tropical region, $-0.10 \pm 0.31$ for the southern mid-latitudes, and $-0.04 \pm 0.14$ % year$^{-1}$ for the southern high-latitudes.

### 3.1 GEOS-Chem tagged simulation

GEOS-Chem (version 9-02: http://acmg.seas.harvard.edu/geos/doc/archive/man.v9-02/index.html, Turner et al., 2015) is a global 3-D CTM capable of simulating global trace gas and aerosol distributions. GEOS-Chem is driven here by assimilated meteorological fields from the Goddard Earth Observing System version 5 (GEOS-5) of the NASA Global Modeling

Assimilation Office (GMAO). The GEOS-5 meteorological data have a temporal frequency of 6 h (3 h for mixing depths and surface properties) and are at a native horizontal resolution of 0.5°x0.667° with 72 hybrid pressure-σ levels describing the atmosphere from the surface up to 0.01 hPa. In the framework of this study, the GEOS-5 fields are degraded for model input to a 2°x2.5° horizontal resolution and 47 vertical levels by collapsing levels above ~80 hPa. GEOS-Chem has been extensively evaluated in the past (van Donkelaar et al., 2012; Park et al., 2006, 2004; Zhang et al., 2011, 2012). These studies show a good

simulation of global transport with no apparent biases.





Emissions for the GEOS-Chem simulations are from the EDGAR v4.2 anthropogenic methane inventory (European Commission, 2011), the wetland model from Kaplan (2002) as implemented by Pickett-Heaps et al. (2011), the GFED3 biomass burning inventory (van der Werf et al., 2010), a termite inventory and soil absorption from Fung et al. (1991), and a biofuel inventory from Yevich and Logan (2003). Wetland emissions vary with local temperature, inundation, and snow cover.

Open fire emissions are specified with 8 hr temporal resolution. Other emissions are assumed aseasonal. Methane loss is mainly by reaction with the OH radical. We use a 3-D archive of monthly average OH concentrations from Park et al. (2004). The resulting atmospheric lifetime of methane is 8.9 years, consistent with the observational constraint of $9.1 \pm 0.9$ years (Prather et al., 2012).

The GEOS-Chem model output presented here covers the period January 2005-December 2012, for which the GEOS-5 meteorological fields are available. We use for this simulation, the best emission inventories available as implemented in version 9-02 of the model and rely on the spatial and temporal distributions of emissions. This tagged simulation includes 11 tracers : one tracer for the soil absorption sink (sa) and ten tracers for sources: gas and oil (ga), coal (co), livestock (li), waste management (wa), biofuels (bf), rice cultivation (ri), biomass burning (bb), wetlands (wl), other natural emissions (on) and

other anthropogenic (oa) emissions. We have used a one-year run for spin-up from January 2004 to December 2004, restarted 70 times for initialization of the tracer concentrations. The model outputs consist of methane mixing ratio profiles saved at a 3-h time frequency and at the closest pixel to each NDACC station. To account for the vertical resolution and sensitivity of the FTIR retrievals, the individual concentration profiles simulated by GEOS-Chem are interpolated onto the FTIR vertical grid (see next section for description of regridding).

**3.1.1 Data regridding and processing**

In order to perform a proper comparison between the GEOS-Chem outputs and our NDACC FTIR observations we accounted for their respective spatial domains and used a conservative regridding scheme so that the total mass of the tracer is preserved (both locally and globally over the entire vertical profile). This was achieved using an algorithm similar to the one described in Sect. 3.1. of Langerock et al. (2015). To this end, time-dependent elevation coordinates are first calculated for the model

outputs using grid-box heights data and providing topography data regridded onto the GEOS-Chem horizontal grid before conservative regridding.

The model outputs (source grid) are then regridded onto an observation-compliant destination grid through our conservative regridding scheme that includes a nearest-neighbor interpolation and a vertical regridding. The vertical destination grid

corresponds to the retrieval grid adopted for each station. Regridded fields (tracer mixing-ratio) may have undefined values, for cells of the destination grid that do not overlap with the model source grid. For grid-cells that partially overlap the model grid, we apply a "mask tolerance", i.e., a relative overlapping volume threshold below which the value of the grid-cell will be set as undefined. This may introduce conservation errors, but since partially overlapping cells are likely to occur only at the top level of the model vertical grid, these errors can be neglected for species that usually have a low mixing-ratio at that level,

such as methane.

To account for the vertical resolution and sensitivity of the FTIR retrievals, the individual concentration profiles simulated by GEOS-Chem are averaged into daily profiles (including day and night simulation) and smoothed according to:

$$x_{smooth} = (x_m - x_a)A + x_a \qquad (1)$$

where A is the FTIR averaging kernels, $x_m$ is the daily mean profile as simulated by the GEOS-Chem model regridded to the observation retrieval grid and $x_a$ the FTIR a priori used in the retrieval according to the formalism of Rodgers [1990]. Averaging





kernels are seasonal averages combining individual matrices from FTIR retrievals. Concerning the methane tracers, we constructed vertical a priori profiles for each of them by scaling the methane a priori employed for each station in order to smooth them as well. To this end, we determined for the ten sites the contribution of each tracer to the total methane on the basis of the mean budget simulated by the model over the 2005-2012 time period.

### 3.1.2 GEOS-Chem simulation vs NDACC FTIR observations

As we previously pointed out, since the information content of the FTIR retrievals prevents from retrieving a pure tropospheric component, we will focus on comparisons between FTIR and GEOS-Chem total columns. Due to the availability of the GEOS-5 meteorological fields and to ensure consistency, we limited our comparison of methane changes between FTIR observations and the GEOS-Chem simulation over the 2005-2012 time period. It is however worth mentioning that methane
changes as observed by our FTIR observations are in agreement for all ten stations (see Fig. 4 and Table 3) between both time periods, i.e. 2005-2012 and 2005-2014.

Firstly, comparisons between FTIR observations and the smoothed GEOS-Chem simulation over the 2005-2012 time period have been performed for each NDACC station, for days when observations are available. Both time series are illustrated on
Fig. 5 as anomalies with respect to 2005.0 (see corresponding reference columns in Table 3). We report a good agreement between FTIR and GEOS-Chem methane with no systematic bias, except for the Tsukuba, Lauder and Arrival Heights stations where GEOS-Chem shows a systematic bias of -3.2 ± 3.1 %, 2.3 ± 1.7 % and 4.8 ± 3.5 % (1-sigma level of uncertainty), with their respective FTIR observations. Since we defined the methane anomaly at 0 % in 2005.0 (or 2006.0 for Eureka) for both our observations and the GEOS-Chem simulation, we consequently corrected this observed bias on Fig. 5. On the other hand,
we observe a slight phase offset between FTIR and GEOS-Chem seasonal cycles for Izaña and Tsukuba. Indeed, GEOS-Chem simulates the maximum methane column 85 days ahead of FTIR measurements for Izaña while it shows a delay of 92 days with respect to the Tsukuba FTIR time series. It should however be pointed out that the seasonal cycle's amplitude is well reproduced by GEOS-Chem with a peak-to-peak amplitude of 5.0 ± 0.9 % for Tsukuba and of 3.6 ± 0.5 % for Izaña while the methane seasonal cycle from FTIR measurements shows a peak-to-peak amplitude of 5.9 ± 1.7 % and 4.3 ± 1.8 %,
respectively.

Regarding the increase of methane as simulated by GEOS-Chem, the simulation indicates a mean annual increase ranging from 0.31 ± 0.03 to 0.43 ± 0.06 %/year and a globally averaged annual change of 0.35 ± 0.03 %/year with respect to 2005.0 (averaged over ten stations, 2-sigma level of uncertainty). Mean annual changes of total columns of methane between 2005
and 2012 for both FTIR measurements and the GEOS-Chem simulation are illustrated on Fig. 4 in blue and orange respectively. In terms of methane increase, the model is in good agreement (within error bars) with the observations except for Jungfraujoch, Izaña and Wollongong where GEOS-Chem shows an overestimation of the methane increase.

We first discuss the possible causes of the slight trend discrepancy between FTIR and GEOS-Chem for Jungfraujoch. At
mountain type sites, subsidence is predominant for anticyclonic weather conditions resulting in adiabatic warming and cloud dissipation. The clear sky and strong radiation conditions lead to the growth of the atmospheric boundary layer (ABL) and thermally induced injections of ABL air can reach the altitude of those sites (Collaud Coen et al., 2011; Nyeki et al., 2000). In addition, mountain venting induced by higher temperatures allows the transport of ABL air to the free troposphere occurring often in summer (between April and August; Henne et al., 2005). At Jungfraujoch, the airmasses originating from the ABL
amount to only 30% of the year (Collaud Coen et al., 2011). More specifically in summer, airmasses originate from the ABL 50% of the time (Collaud Coen et al., 2011). It has been established that vertical export of airmasses above mountainous terrain





is presently poorly represented in global CTMs (Henne et al., 2004). In addition, mean annual changes of methane for summer and winter measurements show that with respectively $0.25 \pm 0.06$ and $0.33 \pm 0.04$ % year$^{-1}$, FTIR measurements and GEOS-Chem agree in summer, during the influence of the ABL, while they do not agree in winter. Indeed, FTIR winter measurements show a non-significant mean annual winter change of $0.10 \pm 0.13$ % year$^{-1}$ while GEOS-Chem shows a mean annual winter

change of $0.23 \pm 0.11$ % year$^{-1}$. Since FTIR measurements and GEOS-Chem agree on the methane changes in summer at Jungfraujoch, when under the influence of the ABL, this seasonal analysis of changes of methane at Jungfraujoch emphasizes the poor representation of summer versus winter thermal convection of air masses from the boundary layer to the free troposphere by the model.

About Izaña, it is worth mentioning that the FTIR methane total column time series shows a smaller seasonal cycle. Indeed, the combination of no local emission sources in the vicinity of Izaña, good mixing of airmasses and a regular solar insolation associated with more constant OH amounts leads to a dampened seasonal cycle (Dlugokencky et al., 1994) at that site. Therefore, small annual changes of methane and smaller uncertainty on the mean annual change computed by the bootstrap method complicates the agreement between the FTIR and GEOS-Chem methane changes. However, as mentioned above, it

should be pointed out that the amplitude of this smaller seasonal cycle is well reproduced by the GEOS-Chem simulation.

About Wollongong, as already pointed out, noisier observations at the beginning of the period of interest may affect the relatively small annual changes of methane overall. In addition, one should not forget that sites such as Izaña or Wollongong can be challenging sites for models to reproduce due to the topography and land-sea contrast (Kulawik et al., 2015).

**3.1.3. Tagged simulation analysis**

The GEOS-Chem tagged simulation, which provides the contribution of each tracer to the total methane simulated, enables us to quantify and express the contribution of each tracer to the global methane increase. In order to do so, we considered year-to-year relative changes according to the following equation:

$$YC \ (in \ \%) = (\mu_n - \mu_{n-1})/\mu_{tot,n-1} \qquad (2)$$

where $\mu_n$ is the annual mean of the simulated methane for the year n. The year-to-year relative changes are computed so that when we assume a relative change of a tracer for the year n, it is expressed with respect to the previous year (n-1) using $\mu_{tot, n-1}$ the annual mean of the simulated cumulative methane for the year (n-1) as a reference. Average of the individual relative year-to-year changes of total methane are in agreement with the mean annual change computed by the bootstrap method within error bars (2-sigma level uncertainty, Table 3). Therefore, the considered relative year-to-year changes of each tracer and for

each site are illustrated on Fig. 6. The first three contributors to the annual methane change over the 2005-2012 time period are displayed for each site in Table A.1 (see Appendix A) along with the cumulative relative increase for the whole 2005-2012 time period.

On a global scale, we observe from the tracer analysis as simulated by GEOS-Chem that natural emission sources such as

emissions from wetlands and biomass burning fluctuate inter-annually and thus are the dominant contributors to the interannual variability in methane surface emissions. This is in agreement with the finding of Bousquet et al. (2011), that fluctuations in wetland emissions are the dominant contribution to interannual variability in surface emissions, explaining 70% of the global emission anomalies over the past two decades, while biomass burning contributes only 15%. Regarding wetlands emissions, the simulation shows a mean net increase of methane in 2006 of +0.30 % (mean value over all sites) attributed to the tracer. In

2007-2008, GEOS-Chem simulates a stabilization of methane in the atmosphere due to the reduction of wetland emissions. Indeed, we observe either a slightly negative change in wetlands methane of $-0.08 \pm 0.07$ % and of $-0.08 \pm 0.04$ % respectively





in 2008 and 2009 (mean values over all sites) or a minor increase not larger than 0.07 % in Arrival Heights (in 2009), in Tsukuba (in 2008) and in the high-latitude sites (i.e. Eureka and Kiruna in 2008 and 2009). On the other hand, the biomass burning tracer globally shows a net increase of 0.10 ± 0.01 % in 2007 likely due to the major fire season in tropical South America (Bloom et al., 2015) and a net decrease of -0.09 ± 0.01 % in 2009 and of -0.07 ± 0.01 % in 2012 with respect to the

previous year. On the sink side, we find a negative phase between the relative year-to-year changes of the soil absorption tracer and the total methane simulated by GEOS-Chem except for Izaña where it remains positive over the time period studied.

On a local scale, we observe a slowdown of the increase in 2010 at mid-latitude sites (i.e. Zugspitze, Jungfraujoch, Toronto) and in 2011 at Tsukuba and at the high-latitude sites of Eureka and Kiruna. Following this stabilization phase, European sites

find a substantial increase of more than 1.15 % in 2011 with respect to the previous year which is mainly due to an anomaly of wetlands emissions (+ 0.38 %) but also as a result of a relative increase of +0.21 % and +0.17 % of emissions from livestock and coal, respectively. The Izaña site presents the most regular increase mainly due to a smaller variability over the whole time period (seasonal cycle of Izaña previously discussed in Sect. 3.1.2.). In contrast, methane over Arrival Heights shows high variability from one year to another, which illustrates how dynamically sensitive the polar air is to transport from lower

latitudes (Strahan et al., 2015).

Finally, regarding anthropogenic emissions, with positive year-to-year changes during the whole 2005-2012 time period, the coal and the gas and oil emissions are both regularly increasing through time. According to the GEOS-Chem tagged simulation, they both rank as the most important anthropogenic contributors to methane changes for all stations (see Appendix A) and thus

substantially contribute to the total methane increase. In fact, the coal and the gas and oil tracers respectively comprise a third (32 %) and almost a fifth (18 %) of the cumulative increase of methane over the 2005-2012 time period while their respective emissions are responsible for only 7.5 and 12.5 % of the methane budget. As a comparison, the cumulative increase of methane emitted from wetlands, amounts to 16 % of the total increase since 2005 while wetland emissions makes up 34 % of the methane budget.

**4. Discussion and conclusions**

The cause of the methane increase since the mid-2000s has been often discussed and has still not been completely resolved (Aydin et al., 2011; Bloom et al., 2010; Dlugokencky et al., 2009; Hausmann et al., 2016; Kirschke et al., 2013; Nisbet et al., 2014; Rigby et al., 2008; Ringeval et al., 2010; Schaefer et al., 2016; Sussmann et al., 2012). On the sink side, Rigby et al. (2008) identified a decrease of OH radicals with a large uncertainty (− 4 ± 14 %) from 2006 to 2007 while Montzka et al.

(2011) found a small drop of ∼1 % year[-1], which might have contributed to the enhanced methane in the atmosphere. On the other hand, Bousquet et al. (2011) reported that the changes in OH remain small (<1% over the 2006-2008 time period). Nevertheless, observations of small inter-annual variations are in agreement with the understanding that perturbations in the atmospheric composition generally buffer the global OH concentrations (Dentener, 2003; Montzka et al., 2011).

The small to non-significant changes of methane in the stratosphere, as reported from analysis of the ACE-FTS methane research product, confirm that the increase of methane takes place in the troposphere. It is indeed driven by increasing sources emitted from the ground (Bousquet et al., 2011; Nisbet et al., 2014; Rigby et al., 2008), affecting primarily its tropospheric abundance and justifying the need for a source-oriented analysis of this recent increase.

Our analysis of the GEOS-Chem tagged simulation determines that secondary contributors to the global budget of methane such as coal mining, gas and oil transport and exploitation, have played a major role in the increase of atmospheric methane



observed since 2005. However, while the simulation we used comprises the best emission inventories available so far, it has its own limitations. Firstly, Schwietzke et al. (2014), Bergamaschi et al. (2013) and Bruhwiler et al. (2014) reported that the EDGAR v4.2 emission inventory overestimates the recent emission growth in Asia. Indeed, Turner et al. (2015) reported from a global GOSAT (Greenhouse gases Observing SATellite) inversion that Chinese methane emissions from coal mining are too

large by a factor of 2. Other regional discrepancies between the EDGAR v4.2 inventory and the GOSAT inversion such as an increase in wetland emissions in South America and an increase in rice emissions in Southeast Asia, have been pointed out by Turner et al. (2015) as well. On the other hand, it has been showed that the current emissions inventories, including EDGAR v4.2, underestimate the emissions of methane associated with the gas and oil use and exploitation, as well as livestock emissions (Franco et al., 2015, 2016; Turner et al., 2015, 2016). Furthermore, Lyon et al. (2016) pointed out that emissions

from oil and gas well pads may be missing from most bottom-up emission inventories. The problem of the source identification clearly resides in the need for a better characterization of anthropogenic emissions and especially in emissions of methane from the oil and gas and livestock sectors.

Concerning the oil and gas emissions, ethane has shown a sharp increase since 2009 of ~5 % year$^{-1}$ at mid-latitudes and of

~3 % year$^{-1}$ at remote sites (Franco et al., 2016) which is attributed to the recent massive growth of oil and gas exploitation in the North American continent, with the geographical origin of these additional emissions confirmed by Helmig et al. (2016). Since ethane shares an anthropogenic source of methane : the production, transport and use of natural gas and the leakage associated to it (at 62 % ; Logan et al., 1981; Rudolph, 1995), Franco et al. (2016) were able to estimate an increase of oil and gas methane emissions ranging from 20 Tg year$^{-1}$ in 2008 to 35 Tg year$^{-1}$ in 2014, using the $C_2H_6/CH_4$ ratio derived from

GOSAT measurements as a proxy, confirming the influence of fossil fuel and gas production emissions impact on the observed methane increase. Moreover, Hausmann et al. (2016) reported an oil and gas contribution to the renewed methane in Zugspitze of 39 % over the 2007-2014 time period based on $C_2H_6/CH_4$ ratio derived from an atmospheric two-box model. However, as Kort et al. (2016) and Peischl et al. (2016) pointed out, the variability in the $C_2H_6/CH_4$ ratio associated to oil and gas production needs to be taken into account in a more rigorous manner as the strength of the $C_2H_6/CH_4$ relationship strongly depends on the

studied region and/or production basin.

In conclusion, we report changes of atmospheric methane between 2005 and 2014 from FTIR measurements performed at 10 ground-based NDACC observation sites for the first time. From the ten NDACC methane time series, we computed a mean global annual increase of total column methane of 0.31 ± 0.03 % year$^{-1}$ (averaged over ten stations, 2-sigma level of

uncertainty), using 2005.0 as reference, which is consistent with methane changes computed from in situ measurements. From the GEOS-Chem tagged simulation, accounting for 11 tracers (10 emission sources and one sink) and covering the 2005-2012 time period, we computed a mean annual change of methane of 0.35 ± 0.03 % year$^{-1}$ since 2005, which is globally in good agreement with the FTIR mean annual changes. In addition, we presented a detailed analysis of the GEOS-Chem tracer changes on both global and local scales over the 2005-2012 time period. To this end, we considered relative year-to-year changes in

order to quantify the contribution of each tracer to the global methane change since 2005. According to the GEOS-Chem tagged simulation, wetland methane contributes mostly to the interannual variability while sources that contribute the most to the observed increase of methane since 2005 are mainly anthropogenic and are coal mining, gas and oil exploitation, and livestock (from largest to smallest contribution). While we showed that GEOS-Chem agrees with our observations and consequently with in situ measurements, the repartition between the different sources of methane would greatly benefit from

an improvement of the global emission inventories. As an example, Turner et al. (2015) suggested that EDGAR v4.2 underestimates the US oil and gas and livestock emissions while overestimating methane emissions associated to coal mining. From the emission source shared by both ethane and methane and from various ethane studies, it is clear that further attention has to be given to improved anthropogenic methane inventories, such as emission inventories associated with fossil fuel and





natural gas production. This is essential in a context of the energy transition that includes the development of shale gas exploitation.

Finally, it is worth mentioning that Schaefer et al. (2016) argue with the fact that thermogenic emissions of methane are responsible for the renewed increase of methane during the mid-2000s. Indeed, from methane isotopologues observations and a one-box model deriving global emission strength and isotopic source signature, Schaefer et al. reports that the recent methane increase is predominantly due to biogenic emission sources such as agriculture and climate-sensitive natural emissions. These results contrast with the context of a booming natural gas production and the resumption of coal mining in Asia. However, it is also worth noting that the $^{13}C/^{12}C$ and D/H ratio of atmospheric methane show distinctive isotope signature depending on the source type (Bergamaschi, 1997; Bergamaschi et al., 1998; Quay et al., 1999; Snover et al., 2000; Whiticar and Schaefer, 2007). In the same way, isotopic fractionation occurs during sink processes with specific ratio depending on the removal pathway (Gierczak et al., 1997; Irion et al., 1996; Saueressig et al., 2001; Snover and Quay, 2000; Tyler et al., 2000). Therefore, the under-exploited analysis of the recent methane increase through trend analysis of methane isotopologues, such as $^{13}CH_4$ and $CH_3D$, is an innovative way of addressing the question of the source(s) responsible for the recent methane increase.

**Acknowledgments**

Most of the data used in this publication were obtained as part of the Network for the Detection of Atmospheric Composition Change (NDACC) and are publicly available (see http://www.ndacc.org). The University of Liège's involvement has primarily been supported by the PRODEX and SSD programs funded by the Belgian Federal Science Policy Office (Belspo), Brussels. The Swiss GAW-CH program is further acknowledged. E. Mahieu is Research Associate with the F.R.S. – FNRS. The F.R.S. – FNRS further supported this work under Grant n° J.0093.15 and the Fédération Wallonie Bruxelles contributed to observational activities support. We thank O. Flock for the constant support during this research. We thank the International Foundation High Altitude Research Stations Jungfraujoch and Gornergrat (HFSJG, Bern) for supporting the facilities needed to perform the observations. The Eureka measurements were made at the Polar Environment Atmospheric Research Laboratory (PEARL) by the Canadian Network for the Detection of Atmospheric Change (CANDAC), led by James R. Drummond and in part by the Canadian Arctic ACE/OSIRIS Validation Campaigns, led by Kaley A. Walker. They were supported by the AIF/NSRIT, CFI, CFCAS, CSA, EC, GOC-IPY, NSERC, NSTP, OIT, PCSP and ORF. Logistical and operational support at Eureka is provided by PEARL Site Manager Pierre Fogal, CANDAC operators, and the EC Weather Station. The Toronto measurements were made at the University of Toronto Atmospheric Observatory (TAO), which has been supported by CFCAS, ABB Bomem, CFI, CSA, EC, NSERC, ORDCF, PREA, and the University of Toronto. We also thank the CANDAC operators, and the many students, postdocs, and interns who have contributed to data acquisition at Eureka and Toronto. Analysis of the Eureka and Toronto NDACC data was supported by the CAFTON project, funded by the Canadian Space Agency's FAST Program. KIT, IMK-ASF would like to thank Uwe Raffalski and Peter Voelgel from the Swedish Institute of Space Physics (IRF) for their continuing support of the NDACC-FTIR site Kiruna. KIT, IMK-ASF would also like to thank E. Sepúlveda for the support in performing the FTIR measurements at Izaña. Garmisch work has been performed as part of the ESA GHG-cci project, and KIT, IMK-IFU acknowledge funding by the EC within the INGOS project. The Centre for Atmospheric Chemistry at the University of Wollongong involvement in this work is funded by Australian Research Council projects DP1601021598 and LE0668470. Measurements and analysis conducted at Lauder, New Zealand and Arrival Heights, Antarctica are supported by NIWA as part of its Government-funded, core research. We thank Antarctica New Zealand for logistical support for the measurements taken at Arrival Heights. A. J. Turner was supported by a Department of Energy (DOE) Computational Science Graduate Fellowship (CSGF). The ACE mission is supported primarily by the Canadian Space Agency. AGAGE is supported principally by NASA (USA) grants to MIT and SIO, and also by DECC (UK) and NOAA (USA) grants to Bristol University





and by CSIRO and the Bureau of Meteorology (Australia). We further thank NOAA for providing in situ data for Alert, Izaña and Halley.

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





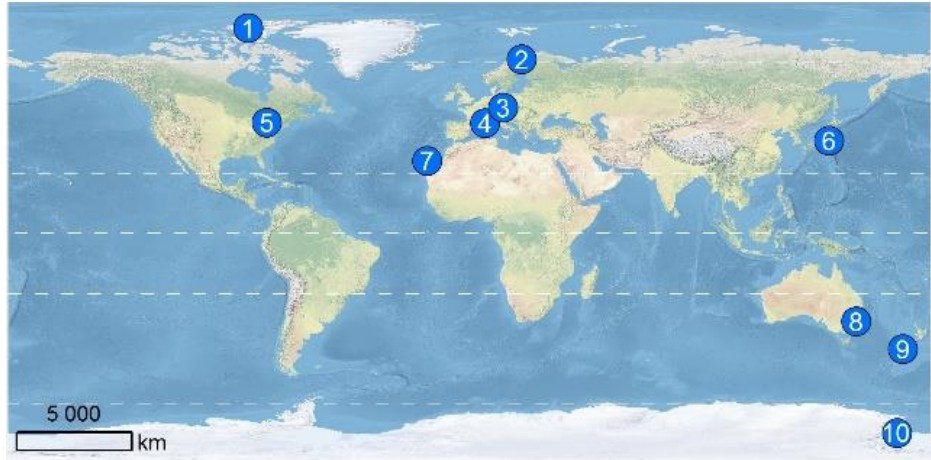

Figure 1: Map of all participating NDACC stations. Detailed coordinates of each station are provided in Table 1.





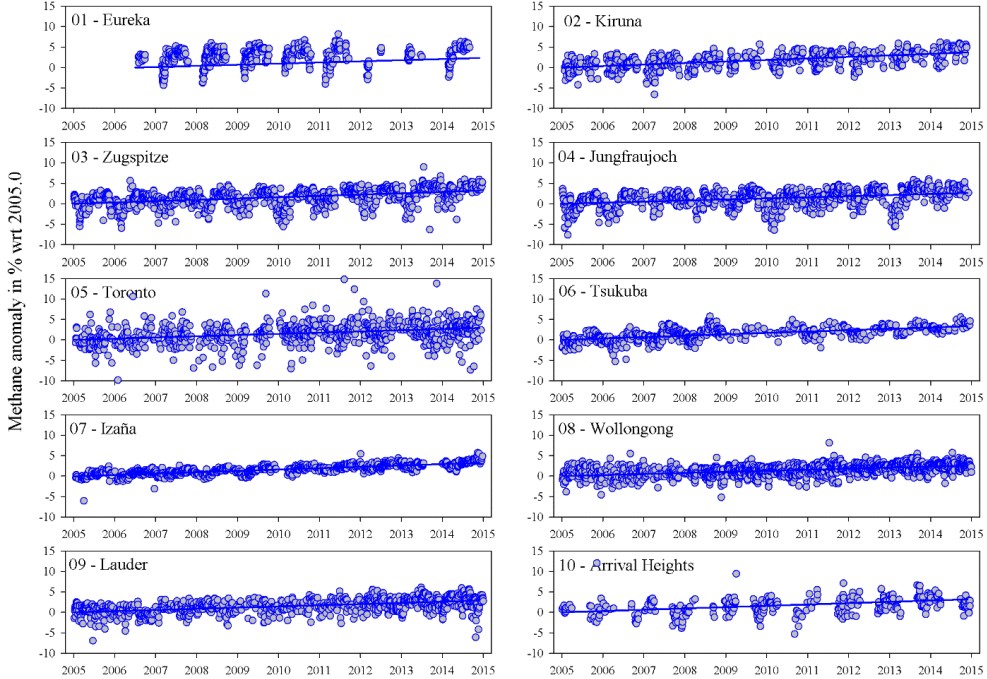

**Figure 2: Daily mean methane anomaly with respect to 2005.0 or 2006.0 (in %) for 10 NDACC stations between 2005 and 2014. The blue line is the linear component of the bootstrap fit (see Sect. 3).**





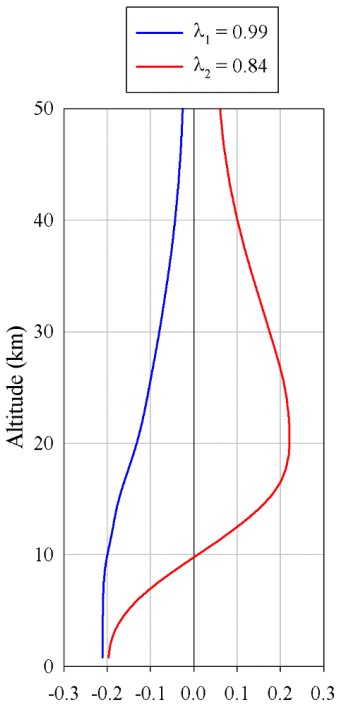

Figure 3: Typical eigenvectors of a NDACC methane retrieval.





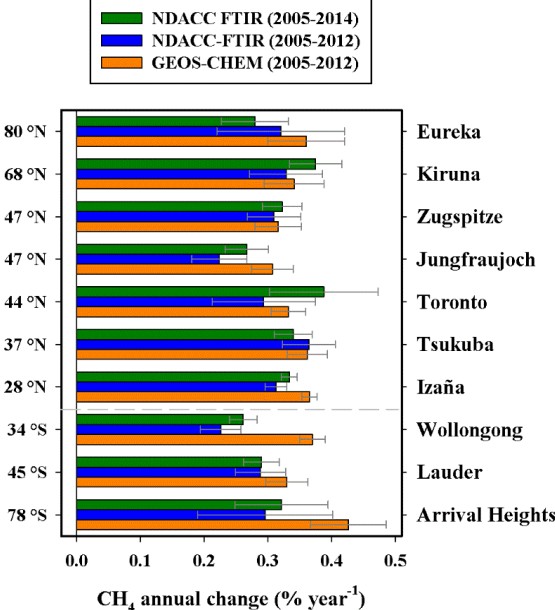

**Figure 4: Methane mean annual change % year⁻¹ with respect to 2005.0 (2006.0 for Eureka), for the FTIR time series between 2005 and 2014 (in blue), the NDACC FTIR time series between 2005 and 2012 (in dark blue), and the GEOS-Chem simulation between 2005 and 2012 (in orange). Grey error bars represent 2-sigma uncertainty.**





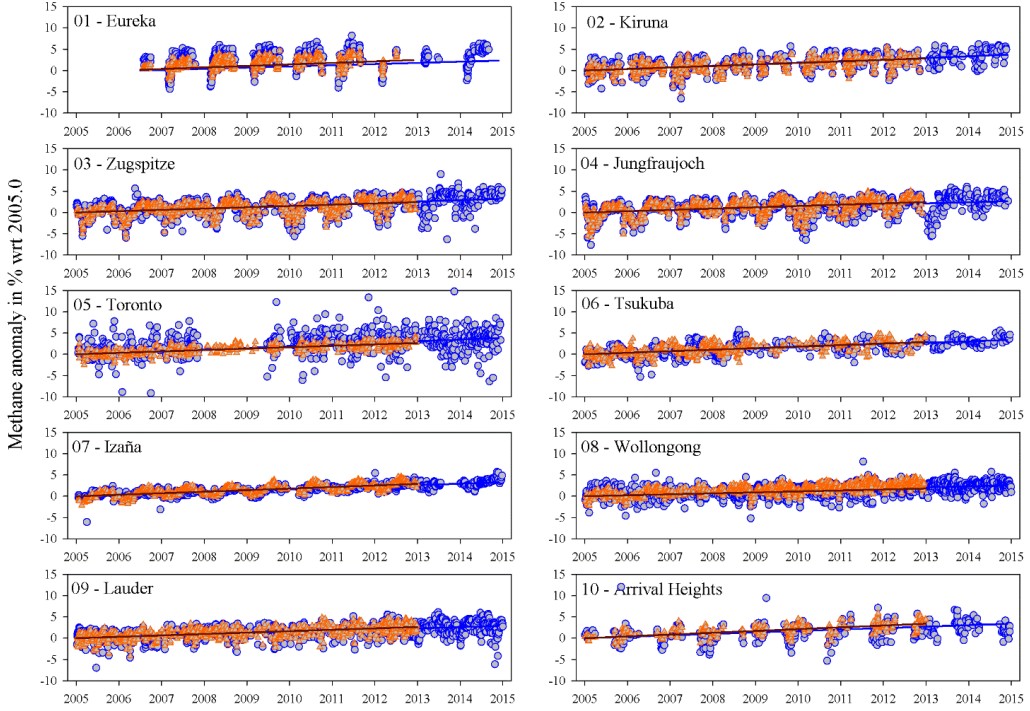

**Figure 5: Daily mean methane anomaly with respect to 2005.0 (in %) for ten NDACC stations between 2005 and 2014 for NDACC FTIR observations (in blue) and between 2005 and 2012 for the smoothed GEOS-Chem simulation (in orange) along with their respective linear component of the bootstrap fit in blue and brown.**





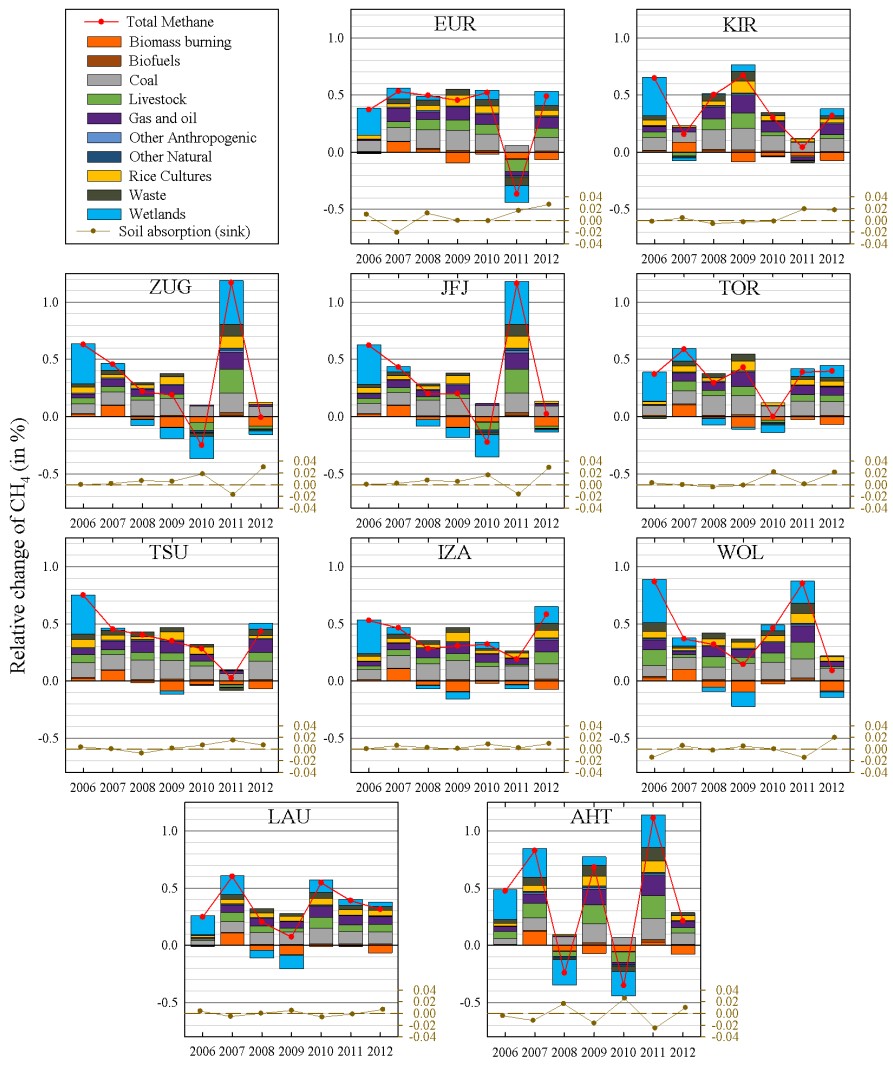

**Figure 6: Year-to-year relative changes in CH₄ due to each emission source (see color codes) for each station (see codes in Table 1) derived from GEOS-Chem. Brown circles represent the year-to-year relative changes of the methane sink due to soil absorption. Red circles illustrate the cumulative year-to-year methane change.**



| Station | Latitude (degrees north) | Longitude (degrees east) | Altitude (m) | # of days[a] | Instrument |
|---|---|---|---|---|---|
| 1 Eureka, EUR (CA) | 80.05 | - 86.42 | 610 | [b]619 | Bruker IFS 125HR |
| 2 Kiruna, KIR (SE) | 67.84 | 20.39 | 420 | 649 | Bruker IFS 120HR |
| | | | | | Bruker IFS 125HR |
| 3 Zugspitze, ZUG (DE) | 47.42 | 10.98 | 2 954 | 1114 | Bruker IFS 125HR |
| 4 Jungfraujoch, JFJ (CH) | 46.55 | 7.98 | 3 580 | 1119 | Bruker IFS 120HR |
| 5 Toronto, TOR (CA) | 43.66 | - 79.4 | 174 | 964 | ABB Bomem DA8 |
| 6 Tsukuba, TSU (JP) | 36.05 | 140.12 | 31 | 640 | Bruker IFS 120HR |
| | | | | | Bruker IFS 125HR |
| 7 Izaña, IZA (ES) | 28.29 | - 16.48 | 2 370 | 990 | Bruker IFS 120M |
| | | | | | Bruker IFS 125HR |
| 8 Wollongong, WOL (AU) | - 34.41 | 150.88 | 31 | 1612 | Bomem DA8 |
| | | | | | Bruker IFS 125HR |
| 9 Lauder, LAU (NZ) | - 45.04 | 169.68 | 370 | 1017 | Bruker IFS 120HR |
| 10 Arrival heights, AHT (NZ) | - 77.83 | 166.65 | 200 | [c]341 | Bruker IFS 120M |

**Table 1: Description of the participating stations. a. Number of days with CH$_4$ measurements available over the 2005-2014 time period. b. Measurements started in 2006 and no measurements between late October and late February due to polar night. c. No measurements between May and August due to polar night.**





| Station | Retrieval code | Retrieval windows | Interfering gases | A priori & regularization | Linelist |
|---|---|---|---|---|---|
| EUR | SFIT-4 | 3[a] | HDO, $H_2O$, $CO_2$, $NO_2$ | [b] WACCM v6 [c] Tikhonov $L_1$ | [d] HIT-08 |
| KIR | PROFFIT | 6[e1] | $H_2O$, HDO, $CO_2$, $O_3$, $N_2O$, $NO_2$, HCl, OCS | WACCM v6 Tikhonov $L_1$ | [f] ad hoc $CH_4$ HIT-08 |
| ZUG | PROFFIT | 3[a] | $H_2O$, HDO, $CO_2$, $NO_2$ | WACCM v6 Tikhonov $L_1$ | [g] HIT-00 |
| JFJ | SFIT-2 v3.94 | 6[e2] | $H_2O$, HDO, $CO_2$, $O_3$, $NO_2$, HCl | WACCM v6 Tikhonov $L_1$ | [f] ad hoc $CH_4$ HIT-08 |
| TOR | SFIT-4 | 3[a] | $H_2O$, HDO, $CO_2$, $NO_2$ | WACCM v6 Tikhonov $L_1$ | HIT-08 |
| TSU | SFIT-2 v3.94 | 3[a] | $H_2O$, HDO, $CO_2$, $NO_2$ | [h] NIES Airplane Tikhonov $L_1$ | HIT-00 |
| IZA | PROFFIT | 6[e1] | $H_2O$, HDO, $CO_2$, $O_3$, $N_2O$, $NO_2$, HCl, OCS | WACCM v6 Tikhonov $L_1$ | [f] ad hoc $CH_4$ HIT-08 |
| WOL | SFIT-2 v3.94 | 3[a] | $H_2O$, HDO, $CO_2$, $NO_2$ | WACCM v5 Tikhonov $L_1$ | HIT-00 |
| LAU | SFIT-2 v3.82 | 3[i] | HDO | [j] URAP at 44°S [k] OEM | HIT-00 |
| AHT | SFIT-2 v3.82 | 3[i] | HDO | [l] ATMOS zonal mean OEM | HIT-00 |

Table 2: Retrieval parameters for each station. a. As detailed in Sussmann et al., (2011) b. Whole Atmosphere Community Climate Model, (Chang et al., 2008). c. Thikonov regularization as detailed in Tikhonov (1963). d. High-resolution transmission molecular absorption database, Hitran-2008 (Rothman et al., 2009). e. (e1) As detailed in or (e2) based on Sepúlveda et al. (2012). f. For all species, Hitran 2008 parameters are used. For methane, ad hoc adjustments performed by KIT, IMK-ASF are used (D. Dubravica, priv. comm., Dec 2012; see also Dubravica et al., 2013). g. Hitran-2000 including 2001 update release (Rothman et al., 2003). h. A priori profile for Tsukuba retrievals include monthly averaged profiles made from airplane measurements over Japan by the National Institute of Environmental Studies, Japan (NIES, http://www.nies.go.jp/index-e.html). i. As detailed in Rinsland et al., (2006). j. A priori profile for Lauder retrieval include annual mean of measurements from the Microwave Limb Sounder (MLS, https://mls.jpl.nasa.gov/) and the Halogen Occultation Experiment (HALOE, http://haloe.gats-inc.com/home/index.php) onboard the Upper Atmosphere Research Satellite (UARS, http://uars.gsfc.nasa.gov/) at 44°S in the framework of the UARS Reference Atmosphere Project (URAP, Grooß and Russell, 2005). k. Optimal Estimation Method based on the formalism of Rodgers (1990). l. A priori for Arrival Heights retrievals include zonal mean of measurements from the Atmospheric Trace Molecule Spectroscopy Experiment (ATMOS) Spacelab 3 over the 14-65 km altitude range (Gunson et al., 1996).



| Unit | FTIR trend [2005-2014] | | FTIR trend [2005-2012] | | FTIR Reference Column | GEOS-Chem trend [2005-2012] | | GEOS-Chem Reference Column |
|---|---|---|---|---|---|---|---|---|
| | x 10$^{16}$ molec cm$^{-2}$ yr$^{-1}$ | % yr$^{-1}$ | x 10$^{16}$ molec cm$^{-2}$yr$^{-1}$ | % yr$^{-1}$ | x 10$^{19}$ molec cm$^{-2}$ | % yr$^{-1}$ | x 10$^{16}$ molec cm$^{-2}$ yr$^{-1}$ | x 10$^{19}$ molec cm$^{-2}$ |
| EUR | 9.54 ± 1.79 | 0.28 ± 0.05 | 10.81 ± 3.47 | 0.32 ± 0.10 | 3.41[a] | 12.35 ± 2.06 | 0.36 ± 0.06 | 3.46 |
| KIR | 13.26 ± 1.46 | 0.37 ± 0.04 | 11.7 ± 2.04 | 0.33 ± 0.06 | 3.54 | 12.04 ± 1.66 | 0.34 ± 0.05 | 3.53 |
| ZUG | 8.33 ± 0.80 | 0.32 ± 0.03 | 7.99 ± 1.09 | 0.31 ± 0.04 | 2.58 | 8.09 ± 9.26 | 0.32 ± 0.04 | 2.56 |
| JFJ | 6.41 ± 0.81 | 0.27 ± 0.03 | 5.39 ± 1.04 | 0.22 ± 0.04 | 2.40 | 7.31 ± 0.78 | 0.31 ± 0.03 | 2.38 |
| TOR | 10.99 ± 3.03 | 0.29 ± 0.08 | 12.85 ± 3.76 | 0.34 ± 0.10 | 3.71 | 12.45 ± 1.01 | 0.33 ± 0.03 | 3.75 |
| TSU | 12.99 ± 1.13 | 0.34 ± 0.03 | 13.90 ± 1.58 | 0.36 ± 0.04 | 3.82 | 13.36 ± 1.17 | 0.36 ± 0.03 | 3.69 |
| IZA | 9.56 ± 0.35 | 0.33 ± 0.01 | 8.96 ± 0.48 | 0.31 ± 0.02 | 2.87 | 10.34 ± 0.34 | 0.36 ± 0.01 | 2.83 |
| WOL | 9.62 ± 0.80 | 0.26 ± 0.02 | 8.33 ± 1.18 | 0.23 ± 0.03 | 3.69 | 13.63 ± 0.74 | 0.37 ± 0.02 | 3.69 |
| LAU | 9.87 ± 0.95 | 0.29 ± 0.03 | 9.81 ± 1.34 | 0.29 ± 0.04 | 3.41 | 11.46 ± 1.15 | 0.33 ± 0.03 | 3.48 |
| AHT | 10.53 ± 2.39 | 0.32 ± 0.07 | 9.70 ± 3.48 | 0.29 ± 0.11 | 3.28 | 14.53 ± 2.02 | 0.43 ± 0.06 | 3.41 |
| Mean | 10.11 ± 2.03 | 0.31 ± 0.03 | 9.94 ± 2.50 | 0.30 ± 0.04 | - | 11.56 ± 2.35 | 0.35 ± 0.03 | - |

Table 3 – Absolute (in molecules cm$^{-2}$ year$^{-1}$) and relative (in % year$^{-1}$) annual change of methane total columns and its associated 2σ-uncertainties from FTIR observations and the GEOS-Chem methane simulation with respect to 2005.0 and to the reference column given in molecules cm$^{-2}$ in the fifth and last columns of this table, respectively. a. Reference column for Eureka is for 2006.0 since no measurements are available before then. The bottom line of the table shows the average of the ten mean annual trends.



## Appendix A

Table A1 illustrates the first three contributors to the annual methane change and their year-to-year changes for each site along with the cumulative relative increase for the whole 2005-2012 time period. The GEOS-Chem tracers are coded as follows: biomass burning (bb), biofuels (bf), coal (co), livestock (li), gas and oil (ga), other anthropogenic sources (oa), other natural sources (on), rice cultivation (ri), waste management (wa), wetlands (wl).

| Station | % | 2005 → 2006 | 2006 → 2007 | 2007 → 2008 | 2008 → 2009 | 2009 → 2010 | 2010 → 2011 | 2011 → 2012 | 2005 → 2012 |
|---|---|---|---|---|---|---|---|---|---|
| **EUR** | tracers | wl 0.23 | co 0.12 | co 0.16 | co 0.17 | co 0.14 | wl -0.15 | wl 0.12 | co 0.87 |
| | | co 0.10 | ga 0.12 | li 0.09 | ga 0.11 | ga 0.11 | li -0.11 | co 0.12 | ga 0.46 |
| | | ri 0.03 | bb 0.09 | ga 0.07 | ri 0.09 | ri 0.09 | wa -0.06 | ga 0.09 | wl 0.42 |
| | **total** | **0.37** | **0.53** | **0.49** | **0.45** | **0.52** | **-0.37** | **0.49** | **2.49** |
| **KIR** | tracers | wl 0.34 | co 0.34 | co 0.17 | co 0.19 | co 0.13 | co 0.09 | co 0.11 | co 0.98 |
| | | co 0.11 | bb 0.11 | ga 0.10 | ga 0.15 | ga 0.09 | bb -0.03 | ga 0.10 | ga 0.51 |
| | | ga 0.05 | ga 0.05 | li 0.09 | ri 0.17 | ri 0.04 | ga -0.03 | wl 0.06 | wl 0.44 |
| | **total** | **0.65** | **0.15** | **0.50** | **0.67** | **0.30** | **0.04** | **0.32** | **2.63** |
| **ZUG** | tracers | wl 0.35 | co 0.11 | co 0.14 | co 0.15 | wl -0.20 | wl 0.38 | co 0.09 | co 0.83 |
| | | co 0.09 | bb 0.10 | ga 0.06 | ga 0.08 | li -0.07 | li 0.21 | bb -0.08 | wl 0.42 |
| | | ri 0.05 | ga 0.07 | li 0.03 | ri 0.07 | bb -0.05 | co 0.17 | *sa 0.03* | ga 0.41 |
| | **total** | **0.63** | **0.46** | **0.22** | **0.19** | **-0.25** | **1.17** | **-0.01** | **2.40** |
| **JFJ** | tracers | wl 0.35 | co 0.11 | co 0.14 | co 0.15 | wl -0.19 | wl 0.38 | wl 0.38 | co 0.09 |
| | | co 0.11 | bb 0.10 | ga 0.06 | ga 0.08 | li -0.06 | li 0.21 | li 0.21 | bb -0.08 |
| | | ri 0.05 | ga 0.06 | li 0.03 | ri 0.07 | bb -0.05 | co 0.17 | co 0.17 | *sa -0.03* |
| | **total** | **0.62** | **0.43** | **0.20** | **0.20** | **-0.22** | **1.16** | **1.16** | **2.41** |
| **TOR** | tracers | wl 0.26 | co 0.12 | co 0.17 | co 0.17 | co 0.09 | co 0.13 | co 0.12 | co 0.87 |
| | | co 0.09 | wl 0.11 | ga 0.07 | ga 0.17 | wl -0.06 | ga 0.08 | wl 0.10 | ga 0.43 |
| | | ri 0.03 | bb 0.10 | li 0.05 | ri 0.08 | bb -0.03 | wl 0.06 | ga 0.07 | wl 0.40 |
| | **total** | **0.37** | **0.59** | **0.29** | **0.43** | **0.00** | **0.39** | **0.40** | **2.46** |
| **TSU** | tracers | wl 0.34 | co 0.13 | co 0.17 | co 0.16 | co 0.12 | co 0.06 | co 0.16 | co 0.94 |
| | | co 0.13 | bb 0.09 | ga 0.10 | ga 0.10 | ri 0.06 | ga 0.02 | ga 0.11 | ga 0.52 |
| | | li 0.07 | ga 0.07 | li 0.07 | ri 0.07 | ga 0.05 | bb -0.03 | li 0.08 | wl 0.39 |
| | **total** | **0.75** | **0.45** | **0.40** | **0.35** | **0.28** | **0.03** | **0.43** | **2.69** |
| **IZA** | tracers | wl 0.30 | co 0.11 | co 0.14 | co 0.16 | co 0.12 | co 0.12 | wl 0.15 | co 0.87 |
| | | co 0.09 | bb 0.11 | ga 0.08 | ga 0.10 | ga 0.07 | ga 0.07 | co 0.13 | ga 0.49 |
| | | ri 0.04 | wl 0.06 | li 0.05 | ri 0.08 | wl 0.06 | ri 0.04 | li 0.11 | wl 0.15 |
| | **total** | **0.53** | **0.46** | **0.28** | **0.31** | **0.32** | **0.19** | **0.58** | **2.67** |
| **WOL** | tracers | wl 0.38 | bb 0.10 | co 0.11 | co 0.15 | co 0.15 | wl 0.19 | co 0.10 | co 0.78 |
| | | li 0.14 | co 0.10 | li 0.09 | ga 0.07 | ga 0.09 | co 0.17 | bb -0.09 | li 0.52 |
| | | co 0.09 | wl 0.07 | ga 0.09 | ri 0.06 | li 0.08 | li 0.15 | ga 0.04 | ga 0.51 |
| | **total** | **0.87** | **0.37** | **0.32** | **0.14** | **0.46** | **0.85** | **0.09** | **3.01** |
| **LAU** | tracers | wl 0.17 | wl 0.17 | co 0.10 | co 0.11 | co 0.13 | co 0.11 | co 0.10 | ca 0.70 |
| | | co 0.04 | bb 0.11 | ga 0.06 | ga 0.06 | wl 0.11 | ga 0.08 | ga 0.07 | ga 0.43 |
| | | ri 0.02 | co 0.09 | li 0.06 | wl -0.12 | li 0.10 | li 0.05 | li 0.07 | li 0.41 |
| | **total** | **0.25** | **0.60** | **0.20** | **0.07** | **0.55** | **0.39** | **0.31** | **2.37** |
| **AHT** | tracers | wl 0.26 | wl 0.25 | wl -0.22 | li 0.17 | wl -0.21 | wl 0.29 | co 0.10 | co 0.75 |
| | | li 0.06 | li 0.13 | bb -0.05 | co 0.17 | li -0.09 | li 0.20 | ga 0.06 | ga 0.48 |
| | | co 0.05 | bb 0.12 | co 0.07 | ga 0.14 | co 0.07 | co 0.18 | li 0.04 | li 0.47 |
| | **total** | **0.47** | **0.83** | **-0.24** | **0.68** | **-0.35** | **1.11** | **0.21** | **2.71** |

**Table A1. Top three simulated tracers contributing the most to the methane changes, per year and per site, in %.**