# Peer review of "The recent increase of atmospheric methane from ten years of ground-based NDACC FTIR observations since 2005"

_Atmospheric Chemistry and Physics, 2016_

## Referee Comment (RC1) · Anonymous Referee #1 · 20 Sep 2016

Review of "Ten years of atmospheric methane from ground-based NDACC FTIR observations" by Bader et al.

This paper determines trends in total column methane from a subset of globally disperse NDACC FTIR sites over a relatively short time period 2005-2014. The data are compared to the GEOS-Chem CTM for trends and via tagged emissions an effort is made to determine year over year changes in methane sources that may contribute to the observed columns.

General / Major Comments

The paper is concise restricting itself to 10 stations and 10 years of data but leaves a considerable amount of similar data out. There are more stations with archived data and many stations have data dating back to the mid 1990's and earlier. A description
of this larger dataset would be considerably more illuminating and reflective of the longevity and efforts of the NDACC network. In a similar context of under reporting, Fig 3 proves stratospheric information is contained in these data yet these trends are not explored. Relative trends in the troposphere vs the stratosphere would be unique and important. Its not clear what advantage Fig 3 delivers when the information is ignored. Instead ACE-FTS date is invoked to discuss any stratospheric trend. While this data record is also long, it is sparse and not as long as the NDACC record. Why was the CTM results stopped at 2013? This appears arbitrary and again leaves out information

Overall the manuscript is very well written. It leaves much analysis from this rich dataset underutilized and/or undeveloped.

There is too little discussion of total uncertainty in the CH4 columns. References to two other papers does not seem adequate in general and in particular given the use of noise in Wollongong data trends later in the paper as a reason for a mismatch in trend (Pg 9). Were errors even used in the trend analysis? If so were then nominal values or real i.e. derived from calculations for each measurement?

Pg 6 Is a discussion of a type of normalization process with a stated purpose "to characterize the possible impact of the choice of the microwindows and spectroscopy on the retrieved methane". A set of data from JFJ is fit by all other stations using their local method and hence a relative station bias to Jungfraujoch is determined and consequently applied to each respective dataset. 1. This does not "characterize" effects of retrieval parameters. There is no further discussion to investigate this issue i.e. there is no actual characterization. There may be valid reasons to use differing parameters at different sites (e.g. interfering species) that may have unintended consequences when fitting JFJ spectra. A characterization exercise might reveal this. 2. The station bias values are not given, only a mean, this further obscures any understanding of the bias. These should at least be listed in Table 2 or 3. 3. Within the NDACC, these types of retrieval parameters are clearly defined yet many stations apparently do no use them (Table 2) this seems counter productive to the goals of the network. 4. Also if the retrievals are not performed to the NDACC standard are they indeed NDACC data? Are these data (meaning the 10y dataset from each station) found on the NDACC Archive? Or are they a separate retrieval? If they are not on the archive then they are not NDACC data and the premise of the manuscript is not at least completely valid.

The process to attain the anomaly plot Fig 5 is not described in sufficient detail. In particular the choice/method of terms in the annual cycles varying by site should be presented.

Pg 13 top paragraph. 'No systematic bias' except of course, for possible biases removed earlier. This 3 station biases mentioned (Tsukuba, Lauder and Arrival Heights) – How are these determined?

Figure 5 is difficult to determine a comparison. Correlation plots would better illustrate the good agreement and bias of the two datasets. These should be replaced.

Minor issues

There were many issues with the model that could be explored. For instance, the seasonal amplitude in AH, Eureka & Kiruna. The inability to reproduce the variability at Toronto. There was mention of, but no explanation for the annual cycle phase difference at Izana & Tsukuba. There is a discussion of annual cycles and some mismatches (eg. JFJ, Izana). These are discussed only qualitatively when the data are available to quantify them. This would be an added benefit for the paper to put this in a table.

Its pointed out that the calculation of the trend at Wollongong could be effected by the early data variability. Could this early data not be removed from the trend calculation and compared as a shorter time series?

Pg 1 Ln 24 it appears misleading to state 'all members of the . . .' The stations are all NDACC stations but not all NDAC station are used.

Pg 1 Ln 36 'secondary contributors' is poorly defined simply minor might be a choice.

Pg1 Ln 41 its not clear what the reference for 0.97 is.

Pg 2 Ln 9 – The statement ‘significant uncertainties’. Is this in a statistical sense? Can these uncertainties actually be stated in the text especially given the discussion of attribution later in the text. Are they known?

Pg 2 Ln 17 What is meant by “global surface climate change”?

Pg 3 Ln 25 Kiruna is not likely the most northern town in Sweden

Pg 4 – It may not be accurate to describe Toronto as a mega city.

Grammar, Spelling, Typographical issues

Pg 8 Ln 17 Fig 5.

Pg 17 Ln 30 shown

Pg 18 Ln 13 sources not tracers.

---

## Referee Comment (RC2) · Anonymous Referee #2 · 20 Sep 2016

This manuscript presents the analysis of atmospheric methane trends derived from FTIR measurements at ten NDACC sites that cover both, Northern and Southern Hemisphere over the period of 2005-2014. Using GEOS-Chem simulations the authors found that - anthropogenic sources of atmospheric CH4 are responsible for the renewed growth of CH4 that has been registered by different observational systems since 2005; - main contributors into the interannual variations of CH4 total columns are the natural sources (wetlands and biomass burning).

General comments:

In Page 5 line 15. The authors noted that CH4 total columns for the Toronto site have a systematic error due to unknown instrument artifact and then made some manipulations with the data which seem to be doubtful. The main issue is how to separate

(in data) the signals, which come from real atmospheric processes and from the instrument that doesn't work in a proper way. Could authors suggest more reasonable criteria/way for the correction of CH4 time series for Toronto? Or, maybe, it would be better to omit Toronto site's data for the period of 2008-2009 from the analysis?

In Page 9 line 34. The explanation of the the lower value of CH4 trend ($\sim$0.22%/yr for 2005-2012) for the Jungfraujoch site given in the paper is in contradiction to the following: - according to a reference (Collaud Coen,2011), the coming of polluted air to the Jungfraujoch site was usually detected using the monitoring of CO, NOx and SO2 concentrations in the ambient air by local sensors. Authors need to bring compelling arguments proving that portions of polluted air, which can reach high altitude site, will significantly influence not only the concentrations of some gases but also the CH4 total column. - for Zugspitze (also a high altitude site), which is located not so far from Jungfraujoch, CH4 trend has the value of 0.31%/yr (2005-2012). Therefore it is worth to explain such noticeable difference between trends for Jungfraujoch and Zugspitze.

In Page 10 line 13. This is not quite clear why "small annual changes of methane and smaller uncertainty ... complicates the agreement between the FTIR and GEOS-Chem ...".

In Page30 Table 3. Methane trends derived from FTIR measurements over 2005-2012 are higher for the stations that are located in the Northern Hemisphere than for sites in the Southern Hemisphere. In comparison to FTIR trends, GEOS-Chem simulations give us an opposite tendency: CH4 trends have lower values for the Northern Hemisphere. Could authors suggest any reasons of such inconsistency between observational and modeling estimations of CH4 trend?

Technical corrections: Table 3. Column "GEOS-Chem trend [2005-2012]", row "Unit": please, check units.

Taking to account the above described considerations the manuscript can be published in ACP.

---

## Referee Comment (RC3) · Anonymous Referee #3 · 3 Oct 2016

Comment on: "Ten years of atmospheric methane from ground-based NDACC FTIR observations"by W. Baader et. al

**General remarks:**

The paper presents 10 years long times series of Methane ($CH_4$) of 10 NDACC sites , which is an actual topic of gran interest. The presented data have a very high quality, is well written and the paper should be published after minor corrections. Specific remarks:

1. **Title:**

   It might be nice to mention the used years "2005-2014" maybe something like: "Ten years of atmospheric methane from ground-based NDACC FTIR observa-

tions between 2005-2014"

2. **Abstract:** Some parts in the abstract could be written more concise and it might be important to mention that the work is based on total vertical **column** measurements.

3. **Retrieval:**

   Some parts of the retrieval description might be done more easier and in a more common way: 1.For most reader a matrix is multiplied from the left side $A \cdot \vec{x}$: Please write the equation instead of the form, of maybe Rodgers 1990: $\mathsf{X}_s mooth = (Xm - xa)\vec{A} + xa$ but $X_s mooth = \vec{A}(Xm - xa) + xa$(1) Rodgers 2000, as everybody else. I admit that is the same, just the AVK matrix id defined in the transposed way $A = A^T$.

4. I do not understand the section 2.2.2 information content and as far as I know is the word *INFORMATION CONTENT* used for the Shannon information measure describing the increase of the information $-log(Det(S_error)/Det(Sa))$ by Rodgers 2000, which is here slightly different used and maybe not really useful but misleading as a title, maybe "Information analysis" might be better.

5. The eigenvector analysis might be an usefull mathematical tool for many applications in OET, like transformation Sa-matrix to the identity . . . , but it might be difficult for the reader and not so easy to be understand in the more general constraint least square fitting approach which includes "Tihkonov"-regularisation. If you want to keep the eigenvectors figure, first of all you have to specify from wich matrix you calculate eigenvectors and in which units you plot it: fraction or VMR. I assume you are doing it from VMR-Averaging kernels and uses VMR. As you work with ten sites I would like to see all of them, to know if this is a more or less harmonic set of retrievals or if you have to be more careful, if altitude dependent CH4 anomalies due to dynamics, stratospheric intrusions .. or other effects will

be seen differently, by different sites. As the ten stations are not harmonized, I would like to see a simple averaging kernel for total column ($AVK_{tot}$) of all ten station, either a typical or an average $AVK_{tot}$. If you will emphasis on the altitude resolved information of the retrieval are important for this study, I would include the mean DOF in one of the tables maybe (Tab. 2), as this gives in addition information on the strength of the constraint at each site and retrieval strategy.

*"established within the NDACC Infrared Working Group that the regularization strength of the methane retrieval strategy should be optimized so that the Degrees of Freedom For Signal (DOFS) is limited to a value of approximately 2 (Sussmann et al., 2011). As a consequence, the typical information content of NDACC methane retrievals will allow us to retrieve tropospheric and stratospheric columns, as displayed in Fig. 3.*, a more common way to look at two partial columns would be to plot the two partial column averaging kernels and the total column averaging kernel.

6. optional: Jungfraujoch and Zugspitze, see the same pattern of annual resolved changes: Especially prominent is the huge difference between 2010 and 2011: That is really interesting and seems consistent for both the MODEL and hopefully also with the FTIR time series. Could you proof this with the FTIR-Model difference? Maybe with a model-control run using an average change, which would result in a MODEL-FTIR residual with a similar structure than the red line in figure 6.

7. Fig 2 is already included in Fig. 5 therefore I would suggest show in Fig 2 the absolute columns not anomalies, either the daily means or even the individual measurements.

8. Table 2: interference species: please defined if the gases in your list are the simultaneously fitted gases or all in addition simulated interference gases, you could replace the column with the interference gases in the main article by DOF

of each stite retrieval strategy. Maybe provide a supplement, where you add more about the 10 retrieval strategies with exact micro windows fitted, prefitted and simulated interference gases.

9. QA/QC: Looking on the time series and the different results of the model, which explain some sites quite well and other less, I wonder if the operators might use different quality selection of spectra and retrieval results and therefore some time series show a higher scatter, as harmonisation of quality control (QC) might be too much effort at this stage, but it may be possible to include just a table which sumarise, the different QC criteria and help the reader to evaluate if only local effects will produce the heterogen image Fig. 5, or maybe also individual quality filters.

10. Table 3: Units column 7 and 8 have to be exchanged: "molec. cm- Yr-1"<->" The $molec.cm^{-2}yr\{-1$ the 2 is missing

---

## Author Comment (AC1) · 26 Dec 2016

From the reviewer's comments and suggestions, we would like to edit the title to : "The recent increase of atmospheric methane from Tten years ofof atmospheric methane from ground-based NDACC FTIR observations since 2005."

In addition, in order to address one of the reviewer's question, we contacted Dr. M. Collaud-Coen who provided substantial information. Therefore, in agreement with Editor Hal Maring, we would like to include her as one of the co-author (affiliation Federal Office of Meteorology and Climatology, MeteoSwiss, 1530 Payerne, Switzerland).

The new list of co-author and affiliations goes as follows :

Whitney Bader 1,2, Benoît Bovy 1, Stephanie Conway 2, Kimberly Strong 2, Dan Smale

[Figure]

3, Alexander J. Turner 4, Thomas Blumenstock 5, Chris Boone 6, Martine Collaud Coen 7, Ancelin Coulon 8, Omaira Garcia 9, David W. T. Griffith 10, Frank Hase 5, Petra Hausmann 11, Nicholas Jones 10, Paul Krummel 12, Isao Murata 13, Isamu Morino 14, Hideaki Nakajima 14, Simon O'Doherty 15, Clare Paton-Walsh 10, John Robinson 3, Rodrigue Sandrin 2, Matthias Schneider 5, Christian Servais 1, Ralf Sussmann 11 and Emmanuel Mahieu 1.

1 Institute of Astrophysics and Geophysics, University of Liège, Liège, Belgium

2 Department of Physics, University of Toronto, Toronto, ON, M5S 1A7, Canada

3 National Institute of Water and Atmospheric Research, NIWA, Lauder, New Zealand

4 School of Engineering and Applied Sciences, Harvard University, Cambridge, MA, USA

5 Karlsruhe Institute of Technology (KIT), Institute of Meteorology and Climate Research (IMK-ASF), Karlsruhe, Germany

6 Department of Chemistry, University of Waterloo, Waterloo, ON, N2L 3G1, Canada

7 Federal Office of Meteorology and Climatology, MeteoSwiss, 1530 Payerne, Switzerland

8 Institute for Atmospheric and Climate Science, ETH Zurich, Zurich, Switzerland

9 Izana Atmospheric Research Centre (IARC), Agencia Estatal de Meteorologia (AEMET), Spain

10 School of Chemistry, University of Wollongong, Australia

11 Karlsruhe Institute of Technology, IMK-IFU, Garmisch-Partenkirchen, Germany

12 CSIRO Oceans & Atmosphere, Aspendale, Victoria, Australia

13 Graduate School of Environment Studies, Tohoku University, Sendai 980-8578, Japan

14 National Institute for Environmental Studies (NIES), Tsukuba, Ibaraki 305-8506, Japan

15 Atmospheric Chemistry Research Group (ACRG), School of Chemistry, University of Bristol, UK

Please see the file attached for the detailed answer to the reviewer's comments.

Please also note the supplement to this comment:
http://www.atmos-chem-phys-discuss.net/acp-2016-699/acp-2016-699-AC1-supplement.zip

---

## Author Response (AR1)

**Author comment on "Ten years of atmospheric methane from ground-based NDACC FTIR observations." by W. Bader et al.**

We would like to thank the referees for their review and constructive comments. Please find below our responses to each comment.

*Referee #1*

*General / Major Comments*

*The paper is concise restricting itself to 10 stations and 10 years of data but leaves a considerable amount of similar data out. There are more stations with archived data and many stations have data dating back to the mid 1990's and earlier. A description of this larger dataset would be considerably more illuminating and reflective of the longevity and efforts of the NDACC network. In a similar context of under reporting, Fig 3 proves stratospheric information is contained in these data yet these trends are not explored. Relative trends in the troposphere vs the stratosphere would be unique and important. It's not clear what advantage Fig 3 delivers when the information is ignored. Instead ACE-FTS date is invoked to discuss any stratospheric trend. While this data record is also long, it is sparse and not as long as the NDACC record.*

We agree that the title may be misleading as to the premise of this paper. We would like to remind that the main objective of this paper is to discuss the recent increase of methane total columns and the possible cause(s) for this still unresolved upturn with the help of a GEOS-Chem tagged simulation, using a suite of sites covering a broad range of latitudes. Following Referee #1's comment and Referee #3's suggestions (see below), the title of this paper will be edited to "The recent increase of atmospheric methane from ten years of ground-based NDACC FTIR observations since 2005." In this framework, while we agree the NDACC data consists of an even largest dataset, the relative trends in the troposphere vs the stratosphere is beyond the scope of this paper.

*Why was the CTM results stopped at 2013? This appears arbitrary and again leaves out information.*

The time period studied for the GEOS-Chem simulation is due to the limited availability of the GEOS-5 meteorological fields. As mentioned in Page 8 Line 10 "The GEOS-Chem model output presented here covers the period January 2005-December 2012, for which the GEOS-5 meteorological fields are available." and in Page 9 Line 7 "Due to the availability of the GEOS-5 meteorological fields and to ensure consistency, we limited our comparison of methane changes between FTIR observations and the GEOS-Chem simulation over the 2005-2012 time period."

*There is too little discussion of total uncertainty in the $CH_4$ columns. References to two other papers does not seem adequate in general and in particular given the use of noise in Wollongong data trends later in the paper as a reason for a mismatch in trend (Pg 9). Were errors even used in the trend analysis? If so were then nominal values or real i.e. derived from calculations for each measurement?*

The bootstrap method used for the trend analysis (Gardiner et al., 2008) accounts for the distribution of the data and by using the residual deviations of a model fit to the data as a representation of the random effects reflected in the data. Through an iterative process (repeated n time, n=5000 in this paper), those residual deviations are included in a new model fit in order to provide a good approximation to the distribution for the trend results. The 2.5 and 97.5 percentiles of this empirical distribution specify a 95% confidence interval associated with the annual change. This 95% confidence interval is mentioned in former Table 3 (now Table 2, see below).

*Pg 6 Is a discussion of a type of normalization process with a stated purpose "to characterize the possible impact of the choice of the microwindows and spectroscopy on the retrieved methane". A set of data from JFJ is fit by all other stations using their local method and hence a relative station bias to Jungfraujoch is determined and consequently applied to each respective dataset.*

The manuscript does not mention any normalization nor correction of each FTIR observations in any way (except for the Toronto retrievals during from 2008 to early 2009). The anomaly mentioned in the paper is computed for displaying purposes, in order for the reader to focus on the observed increase independently from absolute methane total column values which varies from one station to another.

*1. This does not "characterize" effects of retrieval parameters. There is no further discussion to investigate this issue i.e. there is no actual characterization. There may be valid reasons to use differing parameters at different sites (e.g. interfering species) that may have unintended consequences when fitting JFJ spectra. A characterization exercise might reveal this.*
*2. The station bias values are not given, only a mean, this further obscures any understanding of the bias. These should at least be listed in Table 2 or 3.*

In this paper, we use the optimized retrieval strategies currently available. The ideal way of thoroughly characterizing the effects of the retrieval parameters would be to perform an Observation System Simulation Experiment (OSSE) which is beyond the scope of this paper.

Following the reviewer's comment, Page 5, Line 26 has been edited:
> "In order to investigate  on the possible impact of the choice of the microwindows and spectroscopy on the retrieved methane, each strategy has been tested over a set of spectra recorded at the Jungfraujoch station (3068 spectra recorded between 01-01-2005 and 12-31-2012)."

We want to emphasize on how the choice of retrieval strategy has statistically no impact on the computed trends. In addition to the tests performed on a subset of Jungfraujoch observations, a similar exercise has been performed for Lauder observations. Changes of methane in Lauder from the strategy as described in (Sussmann et al., 2011) amounts at 0.28 ± 0.03 %/year which is in agreement with the value of 0.31 ± 0.03 %/year reported in this work. We can therefore conclude that the choice of the retrieval strategy has a marginal impact on methane changes even for wet sites.

*3. Within the NDACC, these types of retrieval parameters are clearly defined yet many stations apparently do no use them (Table 2) this seems counterproductive to the goals of the network.*

Among the official NDACC targets, methane is an exception (see also below), only because improved line parameters are needed in order to fully harmonize the approaches and exploit the spectra. For the other gases, the harmonization is in very good shape, and this comment does not hold true. We would also like to stress that despite this, the systematic bias resulting from the use of slightly different retrieval strategies has only a marginal effect on the total columns, and no effect on the trends.

*4. Also if the retrievals are not performed to the NDACC standard are they indeed NDACC data? Are these data (meaning the 10y dataset from each station) found on the NDACC Archive? Or are they a separate retrieval? If they are not on the archive then they are not NDACC data and the premise of the manuscript is not at least completely valid.*

While the harmonization of the retrieval strategy for methane is still a work under progress (as improved linelist parameters for methane are still required), it is worth mentioning that the retrieval strategies recommended by Sepúlveda et al. (2012) and Sussmann et al. (2011) share their main targeted $CH_4$ absorption lines and constraints. In order to be more comprehensive, Table 2 will be moved to an appendix (now Appendix A) where the limits of the windows and their corresponding interfering species will be mentioned, as well as the averaged DOFS for the time series studied.

The datasets presented here are the current optimized datasets for each station and 90% of them can be found on the NDACC Archive. While the archiving of the data is required for each NDACC site, this process is still under development for some station and mostly depend on funding and manpower availability.

*The process to attain the anomaly plot Fig 5 is not described in sufficient detail. In particular the choice/method of terms in the annual cycles varying by site should be presented.*

There is no adjustment of the seasonal cycle. We call here anomaly the time series expressed with respect to the value of the methane total column in 2005.0 as fitted by the Fourier series respective. For clarity, the equation for the computation of this anomaly is added in the article.

Page 5. Section 2.2.1. "Retrieval strategies"
"The time series produced using the strategies described in Table 2 are illustrated in Fig. 2. In order to better illustrate the observed increase of methane total columns, t̶T̶he various panels show daily mean methane time series expressed as anomalies with respect to a reference column in 2005.0 (2006.0 for the Eureka station), according to the following equation :

$$Anomaly = \frac{C - C_{05}}{(C + C_{05}) \times 1/2} \times 100 \quad (1)$$

where C is the methane total column and $C_{05}$ the methane total column at the time 2005.0 derived from the linear component of the Fourier series (Gardiner et al., 2008) fitted to the time series (see Sect. 3)."

*Pg 13 top paragraph. 'No systematic bias' except of course, for possible biases removed earlier. This 3 station biases mentioned (Tsukuba, Lauder and Arrival Heights) – How are these determined?*

Each bias has been computed with the same method, as mentioned Page 7 Line 27 : "Mean fractional differences are defined as the difference between two datasets divided by their arithmetic average and expressed in percent (see Eq. 2 in Strong et al., 2008)." The same reference has been added in this paragraph. The systematic biases identified between FTIR observations and the GEOS-Chem model have been added to now Table 2 (former Table 3).

*Figure 5 is difficult to determine a comparison. Correlation plots would better illustrate the good agreement and bias of the two datasets. These should be replaced.*

Figure 5 aims at providing a tool for comparison of methane increase since 2005 (and seasonal cycle) as observed and simulated rather than the bias between total column absolute value. In order to compare FTIR and GEOS-Chem methane total columns, the identified systematic biases have been added in Table 2 (former Table 3).

*Minor issues*
*There were many issues with the model that could be explored. For instance, the seasonal amplitude in AH, Eureka & Kiruna. The inability to reproduce the variability at Toronto. There was mention of, but no explanation for the annual cycle phase difference at Izana & Tsukuba. There is a discussion of annual cycles and some mismatches (eg.JFJ, Izana). These are discussed only qualitatively when the data are available to quantify them. This would be an added benefit for the paper to put this in a table.*

While we focused our analysis on comparisons of methane changes since 2005 between the observations and the simulation, quantification of the mismatched seasonal cycle for Tsukuba and Izaña are given in Page 9, Line 25. In order to better appraise the discrepancies or agreement between the FTIR observations and the GEOS-Chem model, the systematic bias between both time series has been added to Table 2 (former Table 3) for each station.

*Its pointed out that the calculation of the trend at Wollongong could be effected by the early data variability. Could this early data not be removed from the trend calculation and compared as a shorter time series?*

As the annual change of methane since 2005 is relatively small, it may easily be affected by noisier observations. However, it's worth mentioning that the best explanation for the discrepancies between FTIR and GEOS-Chem Wollongong methane is that sites such as Izaña or Wollongong "can be challenging sites for models to reproduce due to the topography and land-sea contrast (Kulawik et al., 2015)" (Page 10, Line 22). Removing the early data would only weaken the consistency of our methane changes analysis.

*Pg 1 Ln 24 it appears misleading to state 'all members of the' The stations are all NDACC stations but not all NDAC station are used.*

The sentence has been replaced by "Changes of atmospheric methane total columns ($CH_4$) since 2005 have been evaluated using Fourier Transform Infrared (FTIR) solar observations performed at ten ground-based sites affiliated to the Network for Detection of Atmospheric Composition Change (NDACC)."

*Pg 1 Ln 36 'secondary contributors' is poorly defined simply minor might be a choice.*
We don't think that "minor" is the best choice for non-negligible contributors amounting to 12-15% of the total, and we prefer to stick to the original expression.

*Pg1 Ln 41 its not clear what the reference for 0.97 is.*
The reference (Stocker et al., 2013) has been included.

*Pg 2 Ln 9 – The statement 'significant uncertainties'. Is this in a statistical sense? Can these uncertainties actually be stated in the text especially given the discussion of attribution later in the text. Are they known?*

This is not in a statistical sense as these uncertainties are unknown.

"and it is worth noting that these figures are still affected by significant uncertainties." will be replaced by "although it is worth noting that the global budget of methane remain insufficiently understood."

*Pg 2 Ln 17 What is meant by "global surface climate change"?*
Solomon et al. (2010) focuses on global surface climate change by estimating the contributions of stratospheric water vapor changes to the recent decadal rates of warming through analysis of correlation with the radiative forcing and sea surface temperature decadal changes.

*Pg 3 Ln 25 Kiruna is not likely the most northern town in Sweden*
*Pg 4 – It may not be accurate to describe Toronto as a mega city.*
*Grammar, Spelling, Typographical issues*
*Pg 8 Ln 17 Fig 5.*
*Pg 17 Ln 30 shown*
*Pg 18 Ln 13 sources not tracers*
The text will be modified accordingly.

***Referee #2***
General comments
*In Page 5 line 15. The authors noted that $CH_4$ total columns for the Toronto site have a systematic error due to unknown instrument artifact and then made some manipulations with the data which seem to be doubtful. The main issue is how to separate (in data) the signals, which come from real atmospheric processes and from the instrument that doesn't work in a proper way. Could authors suggest more reasonable criteria/way for the correction of $CH_4$ time series for Toronto? Or, maybe, it would be better to omit Toronto site's data for the period of 2008-2009 from the analysis?*

The time period associated to the instrument artifact of the Toronto data has been carefully identified based on observational logs. The constant offset used to correct this bias has been determined based on a method similar to the one used for the harmonization of two observational sets after a change in the instrumentation, for example. Find below the methane total column time series for Toronto in blue with the biased data displayed in red and the corrected data in green. While leading to the production of a consistent time series for methane observations above Toronto, this bias correction does not significantly impact the trend of methane which remains within the range of the averaged trend of 0.31 ± 0.03 for the 2005-2014 time period.

[Figure]

*In Page 9 line 34. The explanation of the lower value of CH4 trend (0.22%/yr for 2005-2012) for the Jungfraujoch site given in the paper is in contradiction to the following: - according to a reference (Collaud Coen,2011), the coming of polluted air to the Jungfraujoch site was usually detected using the monitoring of CO, NOx and SO2 concentrations in the ambient air by local sensors. Authors need to bring compelling arguments proving that portions of polluted air, which can reach high altitude site, will significantly influence not only the concentrations of some gases but also the CH4 total column. - for Zugspitze (also a high altitude site), which is located not so far from Jungfraujoch, CH4 trend has the value of 0.31%/yr (2005-2012). Therefore it is worth to explain such noticeable difference between trends for Jungfraujoch and Zugspitze.*

A various number of studies describe the Jungfraujoch observation site as a remote site (Henne et al., 2005, 2010; Okamoto and Tanimoto, 2016; Zellweger et al., 2000). Indeed, Ketterer et al., (2014) estimate the PBL height above Kleine Sheidegg (~5 km and 1.5 km horizontal and vertical distance from Jungfraujoch), with the help of remote sensing measurements of windprofiler signal to noise ratio and ceilometer aerosol backscattering profiles and show that influenced by PBL air masses transported upwards during summer. The study by Collaud Coen et al., (2011), through the in situ measurement of aerosol optic parameters, shows that PBL airmasses and its entire chemical and aerosol composition reaches JFJ height mostly during spring and summer as well. Moreover, as mentioned in Okamoto and Tanimoto, (2016), JFJ is exposed mostly to clean free tropospheric air masses in autumn and winter. In late spring and summer, it is intermittently influenced by vertically exported polluted air masses transported in the PBL over Europe (Zellweger et al., 2000, 2003).

On the other hand, Zugspitze has been classified by Henne et al., (2010) as a "weakly influenced, constant deposition" observation site. This difference between JFJ and ZUG while they are only 250 km apart and have a difference in altitude of 600 m can be explained by the more central Alpine location and higher elevation of Jungfraujoch (3580 m a.s.l.) compared to the position and elevation of Zugspitze (2954 m a.s.l.) at the northern flank of the Alps (see topography in the Figure below).

As to the GEOS-Chem model, the ZUG and JFJ simulations are extracted from two nearby pixels (see Figure below) where one is mostly flat (altitude of 781.8 m) and the other one montaneous (1352.1 m high). This explains the closeness of the results. Especially when what distinguishes both stations is mostly the influence of meteorological processes such as thermally driven transport (Forrer et al., 2000; Okamoto and Tanimoto, 2016) occurring at a local scale, which is still poorly represented by CTMs.

[Figure]

Alps topography with altitude (in meters). Red lines represent the GEOS-Chem 2°x2.5° horizontal grid, while red and cyan cross respectively show the Jungfraujoch and Zugspitze location.

From this comment, we would like to edit the mentioned paragraph as follows:

We first discuss the possible causes of the slight trend discrepancy between FTIR observations at Jungfraujoch and Zugspitze as well as with GEOS-Chem for both stations. Indeed, despite their proximity (~250 km apart) and their respective altitude of 3580 m and 2954 m, both Alpine sites show distinct influences from local thermal induced vertical transport. At mountain-type sites, subsidence is predominant for anticyclonic weather conditions resulting in adiabatic warming and cloud dissipation. The clear sky and strong radiation conditions lead to the convective growth of the atmospheric boundary layer (ABL) that  induce thermal injections of ABL air  to the high-altitude observation sites (Collaud Coen et al., 2011; Henne et al., 2005; Nyeki et al., 2000). In addition, mountain venting induced by higher temperatures allows the transport of ABL air to the free troposphere occurring often in summer (between April and August; Henne et al., 2005). While the Jungfraujoch site is a remote site mostly influenced by free tropospheric airmasses with incursions of ABL airmasses during 50% of the sping and summer time (Collaud Coen et al., 2011; Henne et al.,

2005, 2010; Okamoto and Tanimoto, 2016; Zellweger et al., 2000, 2003), the Zugspitze site is more often influenced by the ABL (Henne et al., 2010). For summer, when the influence of the ABL is the largest, the observed changes are in very close agreement, with 0.25 ± 0.06 and 0.26 ± 0.09 %/year$^{-1}$, respectively. Moreover, it has been established that vertical export of airmasses above mountainous terrain is presently poorly represented in global CTMs (Henne et al., 2004). mean annual changes of GEOS-Chem methane agree with the observations in summer, during the influence of the ABL, with 0.33 ± 0.04 and 0.27 ± 0.08 % year$^{-1}$ for Jungfraujoch and Zugspitze respectively.  In contrast, GEOS-Chem shows a mean annual winter change of respectively 0.23 ± 0.11 and 0.19 ± 0.09 % year$^{-1}$ which agrees with Zugspitze change observations but not with Jungfraujoch changes. Since FTIR measurements and GEOS-Chem methane changes comparisons show a disagreement  on the methane changes during winter at Jungfraujoch,  this seasonal analysis of changes of methane at mountaineous observations sites emphasizes the current poor representation of summer versus winter thermal convection of air masses from the boundary layer to the free troposphere by the model.

This question has been addressed as well as the re-writing of this paragraph with the help of Dr. M. Collaud-Coen. Therefore, we would like to add Dr. Collaud-Coen to the list of co-authors.

*In Page 10 line 13. This is not quite clear why "small annual changes of methane and smaller uncertainty ... complicates the agreement between the FTIR and GEOS-Chem...".*

As the changes of methane remain quite small with respect to other atmospheric species (as an example, atmospheric ethane has increased of 4.9 ± 0.91 % year$^{-1}$; Franco et al., 2015), a slight discrepancy between two datasets and associated small uncertainty may easily lead to the conclusion of a disagreement.

*In Page30 Table 3. Methane trends derived from FTIR measurements over 2005-2012 are higher for the stations that are located in the Northern Hemisphere than for sites in the Southern Hemisphere. In comparison to FTIR trends, GEOS-Chem simulations give us an opposite tendency: CH4 trends have lower values for the Northern Hemisphere. Could authors suggest any reasons of such inconsistency between observational and modeling estimations of CH4 trend?*

From Figure 4 of the paper, there is no clear hemispheric bias in methane changes within the 2-σ uncertainty.

[Figure]

**Figure 4: Methane total column mean annual change in % year⁻¹ with respect to 2005.0 (2006.0 for Eureka), for the FTIR time series between 2005 and 2014 (in blue), the NDACC FTIR time series between 2005 and 2012 (in dark blue), and the GEOS-Chem simulation between 2005 and 2012 (in orange). Grey error bars represent 2-sigma uncertainty.**

*Technical corrections:  Table 3.  Column "GEOS-Chem trend [2005-2012]", row "Unit": please, check units.*

The table has been edited accordingly.

***Referee #3***
General remarks:
*1. Title: It might be nice to mention the used years "2005-2014" maybe something like:*
*"Ten years of atmospheric methane from ground-based NDACC FTIR observations between 2005-2014"*

We agree that the title can be improved. Following Referee #1's comment and Referee #3's suggestions, the title of this paper will be edited to "The recent increase of atmospheric methane from ten years of ground-based NDACC FTIR observations since 2005."

*2. Abstract: Some parts in the abstract could be written more concise and it might be important to mention that the work is based on total vertical column measurements.*

The abstract will be edited accordingly :

" Changes of atmospheric methane total columns (CH₄) since 2005 have been evaluated using Fourier Transform Infrared (FTIR) solar observations performed at ten ground-based sites, affiliated to, all members of the Network for Detection of Atmospheric Composition Change (NDACC). From this, we find an increase of atmospheric methane total columns of that amounts to 0.31 ± 0.03 % year⁻¹

(2-sigma level of uncertainty) for the 2005-2014 period. Comparisons with in situ methane measurements at both local and global scales show good agreement. We used the GEOS-Chem Chemical Transport Model tagged simulation that accounts for the contribution of each emission source and one sink in the total methane, simulated over  2005-2012.  After regridding according to NDACC vertical layering using a conservative regridding scheme and smoothing by convolving with respective FTIR seasonal averaging kernels, the GEOS-Chem simulation shows an increase of atmospheric methane total columns of 0.35 ± 0.03 % year$^{-1}$ between 2005 and 2012, which is in agreement with NDACC measurements over the same time period (0.30 ± 0.04 % year$^{-1}$, averaged over ten stations). Analysis of the GEOS-Chem tagged simulation allows us to quantify the contribution of each tracer to the global methane change since 2005. We find that natural sources such as wetlands and biomass burning contribute to the inter-annual variability of methane. However, anthropogenic emissions such as coal mining, and gas and oil transport and exploration, which are mainly emitted in the Northern Hemisphere and act as secondary contributors to the global budget of methane, have played a major role in the increase of atmospheric methane observed since 2005. Based on the GEOS-Chem tagged simulation, we discuss possible cause(s) for the increase of methane since 2005, which is still unexplained."

*3. Retrieval: Some parts of the retrieval description might be done more easier and in a more common way:  1.For most reader a matrix is multiplied from the left side Please write the equation instead of the form, of maybe Rodgers 1990.*
*I admit that is the same, just the AVK matrix is defined in the transposed way A=AT.*

The equation has been modified accordingly.

*4.  I do not understand the section 2.2.2 information content and as far as I know is the word INFORMATION CONTENT used for the Shannon information measure describing the increase of the information Rodgers 2000, which is here slightly different used and maybe not really useful but misleading as a title, maybe "Information analysis" might be better.*
The title has been changed to "Degrees of freedom and vertical sensitivity range."

*5.  The eigenvector analysis might be an useful mathematical tool for many applications in OET, like transformation Sa-matrix to the identity…, but it might be difficult for the reader and not so easy to be understand in the more general constraint least square fitting approach which includes "Tihkonov"-regularisation.  If you want to keep the eigenvectors figure, first of all you have to specify from which matrix you calculate eigenvectors and in which units you plot it: fraction or VMR. I assume you are doing it from VMR-Averaging kernels and uses VMR. As you work with ten sites I would like to see all of them, to know if this is a more or less harmonic set of retrievals or if you have to be more careful, if altitude dependent CH4 anomalies due to dynamics, stratospheric intrusions ..  or other effects will be seen differently, by different sites.  As the ten stations are not harmonized, I would like to see a simple averaging kernel for total column (AVKtot) of all ten station, either a typical or an average AVKtot. If you will emphasis on the altitude resolved information of the retrieval are important for this study, I would include the mean DOF in one of the tables maybe (Tab.  2),  as this gives in addition information on the strength of the constraint at each site and retrieval strategy.*
*"established within the NDACC Infrared Working Group that the regularization strength of the methane retrieval strategy should be optimized so that the Degrees of Freedom For Signal (DOFS) is limited to a value of approximately 2 (Sussmann et al., 2011). As a consequence, the typical*

*information content of NDACC methane retrievals will allow us to retrieve tropospheric and stratospheric columns, as displayed in Fig. 3.", a more common way to look at two partial columns would be to plot the two partial column averaging kernels and the total column averaging kernel.*

As requested, an averaged value of the DOFS along with its associated 1-σ uncertainty has been added to Table A1. The merged-layer kernels, while not included in the new figure as the 10 vectors prevents us to analyze the figure clearly, show for all sites a very similar pattern, with values close to one from the ground/surface up to 20 km. Figure 3 has been edited as follows :

[Figure]

**Figure 3: Typical NDACC methane retrieval. From left to right. First panel : typical individual (blue curves) CH₄ mixing ratio averaging kernels. Second panel : Merged (shades of blue curves) CH₄ mixing ratio averaging kernels. For merged-layer kernels, corresponding atmospheric column are specified in the legend box. Third panel : corresponding two first eigenvectors. Associated eigenvalues are given in the legend.**

*6. optional: Jungfraujoch and Zugspitze, see the same pattern of annual resolved changes: Especially prominent is the huge difference between 2010 and 2011: That is really interesting and seems consistent for both the MODEL and hopefully also with the FTIR time series. Could you proof this with the FTIR-Model difference? Maybe with a model-control run using an average change, which would result in a MODEL-FTIR residual with a similar structure than the red line in figure 6.*

The proximity of both Jungfraujoch and Zugspitze stations (~250 km away and 600m altitude difference) explains how GEOS-Chem simulates similar yearly changes. However, as stated above, CTMs poorly represent processes occurring at a local scale such as thermally driven transport (Forrer et al., 2000; Okamoto and Tanimoto, 2016), the cause of the differences between observed changes at both stations.

*7. Fig 2 is already included in Fig. 5 therefore I would suggest show in Fig 2 the absolute columns not anomalies, either the daily means or even the individual measurements.*

As methane total columns cover different range of values from one station to another (mostly depending on the altitude of the station), we would like to keep the current version of the figure in order to keep homogeneous axes from one panel to another.

*8.  Table 2:  interference species:  please defined if the gases in your list are the simultaneously fitted gases  or  all  in  addition  simulated  interference  gases,  you  could  replace  the  column  with  the interference gases in the main article by DOF of  each  site  retrieval  strategy.   Maybe  provide  a supplement,  where  you  add more about the 10 retrieval strategies with exact micro windows fitted, prefitted and simulated interference gases.*

Following the reviewer suggestion, Table 2 will be moved to an appendix (now Appendix A) where the limits of the windows and their corresponding interfering species will be mentioned. The averaged DOFS and their associated 2σ-uncertainties for each dataset have been added.

*9.  QA/QC: Looking on the time series and the different results of the model, which explain some sites quite well and other less, I wonder if the operators might use different quality selection of spectra and retrieval results and therefore some time series show a higher scatter, as harmonisation of quality control (QC) might be too much effort at this stage, but it may be possible to include just a table which sumarise, the different QC criteria and help the reader to evaluate if only local effects will produce the heterogen image Fig. 5, or maybe also individual quality filters.*

That is correct, QC is not yet harmonized in NDACC. However, due to different latitude, altitude, humidity and instrumental performances, signal to noise of the data will still be different, differences that are captured by the error estimates.

From the ten time series illustrated on Figure 5, higher scattered data is more pronounced for urban sites such as Toronto and Wollongong. The scatter is mainly due to local pollutions and different wind direction rather than due to quality check of the spectra. For instance, the anomaly for Toronto is up to 15 % which is beyond the error estimate.

However, it is important here to emphasize on the fact that the bootstrap resampling method minimizes the impact of the scattered data on the computed methane changes between 2005 and 2014.

*10.  Table 3: Units column 7 and 8 have to be exchanged: "molec. cm- Yr-1"<->" The molec.cm −2yr {−1 the 2 is missing.*
The manuscript will be modified accordingly.

[revised manuscript text omitted]

**Relevant changes**

1- From the reviewer's comments and suggestions, we would like to edit the title to : "The recent increase of atmospheric methane from Ten years of atmospheric methane from ground-based NDACC FTIR observations since 2005."

2- In addition, in order to address one of the reviewer's question, we contacted Dr. M. Collaud-Coen who provided substantial information. Therefore, in agreement with Editor Hal Maring, we would like to include her as one of the co-author (affiliation Federal Office of Meteorology and Climatology, MeteoSwiss, 1530 Payerne, Switzerland).

The new list of co-author and affiliations goes as follows :

[revised manuscript text omitted]

Table 2: Retrieval parameters for each station. a. As detailed in Sussmann et al., (2011) b. Whole Atmosphere Community Climate Model, (Chang et al., 2008). c. Thikonov regularization as detailed in Tikhonov (1963). d. High-resolution transmission molecular absorption database, Hitran-2008 (Rothman et al., 2009). e. (e1) As detailed in or (e2) based on Sepúlveda et al. (2012). f. For all species, Hitran 2008 parameters are used. For methane, ad hoc adjustments performed by KIT, IMK-ASF are used (D. Dubravica, priv. comm., Dec 2012; see also Dubravica et al., 2013). g. Hitran-2000 including 2001 update release (Rothman et al., 2003). h. A priori profile for Tsukuba retrievals include monthly averaged profiles made from airplane measurements over Japan by the National Institute of Environmental Studies, Japan (NIES, http://www.nies.go.jp/index-e.html). i. As detailed in Rinsland et al., (2006). j. A priori profile for Lauder retrieval include annual mean of measurements from the Microwave Limb Sounder (MLS, https://mls.jpl.nasa.gov/) and the Halogen Occultation Experiment (HALOE, http://haloe.gats-inc.com/home/index.php) onboard the Upper Atmosphere Research Satellite (UARS, http://uars.gsfc.nasa.gov/) at 44°S in the framework of the UARS Reference Atmosphere Project (URAP, Grooß and Russell, 2005). k. Optimal Estimation Method based on the formalism of Rodgers (1990). l. A priori for Arrival Heights retrievals include zonal mean of measurements from the Atmospheric Trace Molecule Spectroscopy Experiment (ATMOS) Spacelab 3 over the 14-65 km altitude range (Gunson et al., 1996).

| Unit | FTIR trend [2005-2014] | | FTIR trend [2005-2012] | | FTIR Reference Column | Bias | GEOS-Chem trend [2005-2012] | | GEOS-Chem Reference Column |
|---|---|---|---|---|---|---|---|---|---|
| | $\times 10^{16}$ molec cm$^{-2}$ yr$^{-1}$ | % yr$^{-1}$ | $\times 10^{16}$ molec cm$^{-2}$yr$^{-1}$ | % yr$^{-1}$ | $\times 10^{19}$ molec cm$^{-2}$ | % | $\times 10^{16}$ molec cm$^{-2}$ yr$^{-1}$ | % yr$^{-1}$ | $\times 10^{19}$ 
[revised manuscript text omitted]